

# Evaluating state-of-the-art process-based and data-driven models in simulating CO₂ fluxes and their relationship with climate in western European temperate forests

Gaïa Michel[1,2], Julien Crétat[1], Olivier Mathieu[1], Mathieu Thévenot[1], Andrey Dara[3], Robert Granat[3], Zhendong Wu[4,5], Clément Bonnefoy-Claudet[1], Julianne Capelle[1], Jean Cacot[6], John S. Kimball[7]

[1] Biogéosciences, UMR 6282 CNRS, Université de Bourgogne, Dijon, France
[2] AgroParisTech, 91120, Palaiseau, France
[3] CarbonSpace Ltd., D04H1F3 Dublin, Ireland
[4] Department of Physical Geography and Ecosystem Science, Lund University, Lund, Sweden
[5] ICOS ERIC, Carbon Portal, Lund, Sweden
[6] Centre National de la Propriété Forestière, Saulieu, France
[7] Numerical Terradynamic Simulation Group, University of Montana, Missoula, MT, United States of America

*Correspondence to*: Julien Crétat (julien.cretat@u-bourgogne.fr)



**Abstract.**
This study evaluates two process-based (LPJ-GUESS and SMAP-L4C) and two data-driven (CarbonSpace and
FLUXCOM) models to capture the temporal variability of $CO_2$ flux exchanges (GPP, RECO and NEE) of
evergreen needleleaf and deciduous broadleaf forests (ENFs and DBFs) in temperate western Europe and its
relationship with climate. Three sites from the FLUXNET network are considered together with two non-
instrumented sites located in Burgundy (North-East France). The focus is put on the representation of the annual
cycle, annual budget, interannual variability and "long-term" trend. The data-driven models are the best models
for representing the mean annual cycle and mean annual budget in $CO_2$ fluxes despite magnitude uncertainties. In
particular, the models accounting for plant functional types in their outputs tend to simulate more marked annual
cycle and lower annual $CO_2$ sequestration for DBFs than ENFs in Burgundy. At the interannual timescale, the $CO_2$
flux – climate relationship is stronger for GPP and RECO than NEE, with increased $CO_2$ fluxes when 2 m
temperature, vapor pressure deficit and evapotranspiration increase and when precipitation and soil moisture
decrease. The models forced by dynamic climate conditions clearly outperform those driven by static climate
conditions. The "long-term" trend is not obvious for NEE neither in the observations nor in the simulations, partly
because both GPP and RECO tend to increase in western Europe. Our results suggest that the spatial resolution of
the climate drivers is likely very important for capturing spatial and temporal patterns in $CO_2$ exchanges and point
towards the need to choose the appropriate model and spatial resolution according to the scientific question to deal
with.

**Key words:** Net ecosystem exchange, gross primary production, ecosystem respiration, climate, annual cycle,
annual budget, interannual variability, trend



## 1 Introduction

Among all their environmental benefits, forest ecosystems are efficient carbon sinks and constitute a potential lever for climate change mitigation. At the global scale, forest ecosystems cover about 30% of landmasses. They represent the largest part of the land carbon sink (Lindeskog et al., 2021), with up to 20-50% of anthropogenic $CO_2$ emissions (land-use changes excluded) sequestered for the 2000-2010 period (Pan et al., 2011; Le Quéré et al., 2018; Pugh et al., 2019). Despite the fertilization effect of increased atmospheric $CO_2$ concentrations (Walker et al., 2020; IPCC, 2023) and the warming-induced lengthening of the growing season (Prislan et al., 2019; Menzel et al., 2020; IPCC, 2023), the evolution in the net ecosystem exchange (NEE) suggests a recent decrease of annual $CO_2$ storage in forest ecosystems of temperate Europe, due to severe heat waves and droughts that affected Northern regions in 2018 and Central-Southeastern regions in 2020 (Smith et al., 2020; Thompson et al., 2020; van der Woude et al., 2023). This trend results from a combination of multiple factors. In France, for instance, the $CO_2$ storage by forests dropped from ~53 Mt $CO_2$ year$^{-1}$ to ~32Mt $CO_2$ year$^{-1}$ between 2005-2013 and 2012-2020, mostly due to increased timber-extraction (+20%), climate-related mortality (+54%) and decreased biological production (-10%) (IGN, 2022; Chuine et al., 2023). Such a continental-to-country scale evolution of forest-related $CO_2$ fluxes needs to be refined at a finer spatial grain to better account for the contributing influence of different forest stands and to clarify the role of forest ecosystems in the $CO_2$ budget at a territorial level and their leverage in mitigating climate change impacts.

A territorial-scale assessment remains, however, challenging. Measuring NEE and its two components, gross primary production (GPP) quantifying $CO_2$ sequestration through photosynthesis and ecosystem respiration (RECO) releasing $CO_2$ through autotrophic and heterotrophic processes, is expensive since it requires the installation and maintenance of flux towers measuring eddy covariance above the canopy (Burba, 2021). The FLUXNET initiative provides over 1500 site-years of quality-controlled flux tower data from 212 sites around the globe, using the same ONEFlux processing pipeline to foster inter-site comparisons (Pastorello, 2020). At the European scale, the Integrated Carbon Observation System (ICOS) network provides standardized and open data from 98 ecosystem stations across 16 countries. The flux towers remain limited in number and unevenly distributed spatially, which makes it impossible to study $CO_2$ fluxes directly in unequipped sites. Process-based and data-driven models allow us to tackle the above limitation. Process-based models, such as dynamical vegetation models, are routinely used to assess $CO_2$ flux exchanges between the atmosphere and the biosphere (Friedlingstein et al., 2023). These are mechanistic models (Friedlingstein et al., 2006; Sitch et al., 2008), which allow for testing the response of $CO_2$ fluxes to individual and combined forcing. Data-driven models rely on the identification of statistical relationships between flux tower measures by eddy-covariance and corresponding land use, vegetation properties and climate characteristics. Based on these statistical relationships, empirical models are built and used for upscaling, i.e., for assessing $CO_2$ fluxes in regions where they are not measured (Jung et al., 2009, 2019, 2020; Tramontana et al., 2016; Zhuravlev et al., 2022). Both approaches have limitations. For instance, estimations of $CO_2$ flux exchanges are highly sensitive to physical parameterizations (Cai and Prentice, 2020) and atmospheric forcing (Wu et al., 2017; Hardouin et al., 2022) in process-based models. Regional $CO_2$ flux upscaling methods are also limited by the sparse and uneven distribution of flux tower measurements, and limitations of the underlying statistical methods used in data-driven models (Jung et al., 2020).



This study aims at comparing the respective strengths and limitations of process-based and data-driven approaches
to capture the recent temporal dynamics of $CO_2$ flux exchanges observed in western European temperate forest
ecosystems, with a focus on evergreen needleleaf forests (ENFs) and deciduous broadleaf forests (DBFs). The first
objective is to discuss their capability to simulate the mean state, interannual variability and trend in NEE and,
when available, GPP and RECO. Previous observation-based studies have shown that $CO_2$ flux exchanges depend
on multiple factors not necessarily related to climate such as soil properties (Kurbatova et al., 2008; Besnard et al.,
2018; Curtis and Gough, 2018; Martinez del Castillo et al., 2022), forest management practices (Carrara et al.,
2003; Scott et al., 2004; Saunders et al., 2012), tree age (Kurbatova et al., 2008; Besnard et al., 2018; Chuine et
al., 2023) and tree species (Carrara et al., 2003, 2004; Welp et al., 2007; von Buttlar et al., 2018; Zheng et al.,
2021; Kong et al., 2022) among many others. On average, the annual cycle of $CO_2$ flux exchanges significantly
differs between ENFs and DBFs since photosynthesis can occur all year long in the former, while is bounded from
spring (bud break) to fall (leaf senescence) in the latter. As a result, DBFs tend to be a net $CO_2$ sink during the
warm season, and $CO_2$ source during the cold season (Granier et al., 2002; Welp et al., 2007); whereas, ENFs can
persist as a $CO_2$ sink year-around under favorable meteorological conditions (Mizoguchi et al., 2012). At the
interannual timescale, Welp et al. (2007) found that the NEE variability is greater and mainly driven by GPP in
Alaskan DBFs and by RECO in the ENFs. This is at odds with Yuan et al. (2009) who found the opposite pattern
in 30 northern-hemisphere sites, suggesting latitudinal (hence climate) dependency in the results.
The second study objective is to examine the influence of climate on the temporal variability of $CO_2$ flux exchanges
in temperate DBFs and ENFs in terms of annual cycle (monthly timescale), interannual variability (monthly and
annual timescales) and trend (annual timescale). The recent record-breaking temperatures and long drought
episodes observed e.g., in Central Europe in 2003, Central and Northern Europe in 2018 and Central and
Southeastern Europe in 2022, have been accompanied by sharp reductions in forest $CO_2$ uptake (Ciais et al., 2005;
Thompson et al., 2020; van der Woude et al., 2023). Understanding the role of climate on forest NEE temporal
dynamics requires accounting for both monthly and annual budgets since potential compensations of $CO_2$ fluxes
can occur across the annual cycle. This is the case in 2018 in Northern Europe when increased $CO_2$ uptake in
spring (due to anomalously warm conditions) was offset by an anomalous decrease in summer (due to heat and
drought), resulting in week NEE anomalies at the annual timescale (Thompson et al., 2020). Understanding the
role of climate on NEE also requires assessing how the much larger GPP and RECO component fluxes may
respond differently to climate. The annual cycle and, to a lesser extent, the interannual variability of these $CO_2$
fluxes are driven by temperature and the water cycle, including soil moisture (Haszpra et al., 2005; Tang et al.,
2013; Kong et al., 2022; Sharma et al., 2022; Li et al., 2023). Welp et al. (2007) showed that DBFs are more
sensitive to soil moisture changes in ENFs than in DBFs, and that decreased GPP under water stress was observed
in DBFs only. The authors attributed this difference to a possible buffer effect in ENFs' soils that is damping out
temperature increases and to a lower stomatal sensitivity of conifers. In addition, the soil respiration increases
exponentially with temperature (van't Hoff, 1898; Meyer et al., 2018) until a maximum temperature threshold is
reached, which rarely occurs in extratropical soils (von Buttlar et al., 2018). However, when extreme temperatures
are combined with soil water stress, clearer GPP and RECO answers come out. For instance, Ciais et al. (2005)
estimated a 30% decrease in GPP and moderate RECO tail-off during the 2003 severe heat and drought event in
Central Europe, resulting in a lower net carbon uptake. The larger contribution of GPP on NEE interannual
variability remains site and stand dependent (Welp et al., 2007; Yuan et al., 2009). Finally, despite strong effects



of recent heat waves and droughts, the NEE does not always show clear trends in response to recent and projected
climate change (Ahlström et al., 2012; Abdalla et al., 2013; Tang et al., 2013; Kong et al., 2022; Martinez del
Castillo et al., 2022; Li et al., 2023). One possible hypothesis, tested in our study, is a potential compensation of
trends between GPP and RECO.
The novelties of this study rely on (i) the comparison between two data-driven models providing $CO_2$ flux
estimations either globally but at coarse resolution (0.5° x 0.5°) or locally but at the hectometric resolution and (ii)
the inclusion of a newly released process-based model constrained by soil moisture satellite data, which provides
$CO_2$ flux estimations for each plant functional type at relatively high space-time resolution (daily; 9 km mesh with
1 km sub-grids). Another originality relies on the multi-scale (annual cycle, interannual variability and trend)
assessment of the temporal variability in estimated NEE (and its two components) and its climate drivers.
The paper is structured as follows. Section 2 presents the materials and methods. Section 3 presents our results at
the monthly and annual timescales and Sections 4 and 5 discuss the results and give the main conclusions,
respectively.
**2 Materials & Methods**

**2.1 Site description**

This study focuses on five forest sites: two non-instrumented sites in northeastern France where NEE, GPP and
RECO are simulated by process-based and data-driven models, and three sites from the FLUXNET network where
NEE is measured and GPP and RECO are calculated (Fig. 1).

The first non-instrumented site is located in the National Park of Forests, a 240,000 ha park mostly covered by
DBFs (50%). One DBF plot of 25 ha is selected because soil respiration measures are conducted there by the
Biogéosciences laboratory since 2020. This DBF plot, named "Châtillonnais (DBF)" hereafter, is located on a
~380 m plateau and characterized by uneven-aged and mixed DBFs dominated by beech (*Fagus sylvatica)* and
oaks (*Quercus robur, Quercus petraea*) with no sylvicultural interventions for ~30 years and by oolitic limestone
soils. The second site is located in the Regional Natural Park of Morvan, on the Mont Beuvray, a semi-mountainous
domain of 950 ha peaking at 821 m and sitting on volcanic-sedimentary rocks. The Mont Beuvray location is
particularly impacted by climate change (Castel et al., 2019), with a mean warming trend reaching 2°C more than
the neighboring lowlands over the 1958-2015 period. Two plots are considered for Mont Beuvray: one even-aged
large-sized Douglas fir *(Pseudotsuga menziesii)* plots of 15 ha classified as ENF and one even-aged beech plot
with continuous cover of 8 ha classified as DBF. These plots are named "Mont Beuvray (DBF)" and "Mont
Beuvray (ENF)" hereafter.



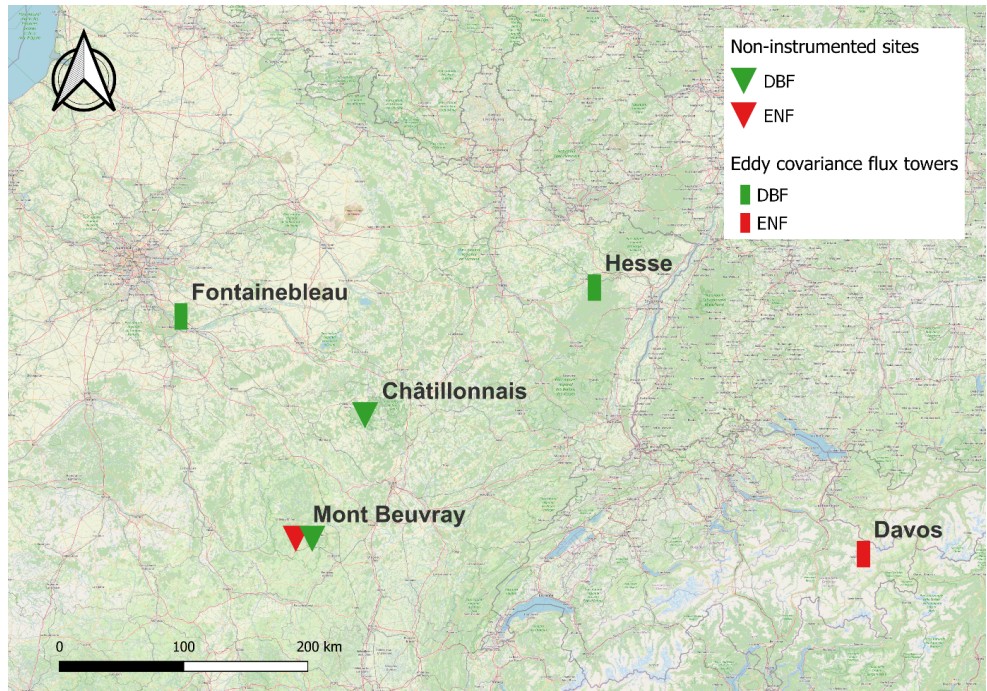

**Figure 1:** Location of the study sites. The rectangles correspond to FLUXNET sites where $CO_2$ fluxes are measured by eddy covariance flux towers and estimated by process-based and data-driven models. The triangles correspond to non-instrumented sites where $CO_2$ fluxes are estimated by process-based and data-driven models only. Symbols in green and red correspond to DBF and ENF sites, respectively. © OpenStreetMap contributors 2021. Distributed under the Open Data Commons Open Database License (ODbL) v1.0.

To compare the $CO_2$ flux dynamics of these sites and to evaluate the accuracy of data-driven and process-based models, we selected three forest tower sites from the FLUXNET network for their resemblance to the aforementioned ones in terms of location, climate or stand characteristics:

- Two lowland DBFs. The "Fontainebleau" site (FR-Fon) is located in the domanial forest of Barbeau (southeast of Paris), dominated by oak (*Qu. petraea*) and characterized by a loamy soil on top of burstones and deeper marls. The "Hesse" site (FR-Hes) is located in the plain east of the Vosges mountains, dominated by beech (*Fagus sylvatica*) and characterized by a deep silty clay soil on sandstone;
- One midland ENF, "Davos" (CH-Dav), located in the middle range of the subalpine belt in the eastern part of the Swiss Alps at 1639 m, dominated by Norway spruce (*Picea abies*) and characterized by a thin soil on schists and gneiss.



**2.2 Carbon flux data**

Measured $CO_2$ fluxes are used as a reference to evaluate outputs from two data-driven and two process-based
models (Table 1). They come from the Warm Winter 2020 release (Warm Winter 2020 Team and ICOS Ecosystem
Thematic Centre, 2022), an update of the FLUXNET2015 dataset (Pastorello et al., 2020) available on the ICOS
platform (https://www.icos-cp.eu/data-products). For each site, we selected daily time series of NEE
(NEE_VUT_REF) accounting for multiple friction velocity thresholds and associated with a favorable quality
control flag above 80%, GPP (GPP_DT_VUT_REF) and RECO (RECO_DT_VUT_REF). GPP and, to a lesser
extent, RECO are less sensitive to the partitioning method (Fig. A1) and the climate – $CO_2$ flux relationship is
similar regardless of the partitioning method used. Here, we retained those $CO_2$ flux data derived from the daytime
flux partitioning method (Lasslop et al., 2010). The temporal coverage of the data is site dependent: 7 years for
Hesse, 18 for Fontainebleau and 24 for Davos (Table 1).

|  | Observations | Process-based models | | Data-driven models | |
|---|---|---|---|---|---|
|  | FLUXNET | SMAP-L4C | LPJ-GUESS | CarbonSpace | FluxCom |
| Parameters | NEE, GPP, RECO, weather | NEE, GPP, RECO | NEE, GPP, RECO | NEE | NEE, GPP, RECO |
| Timescale | Daily | Daily | Hourly | Monthly | Monthly |
| Spatial resolution | Local | 9 km | 50 km | Hectometric | 50 km |
| Temporal coverage | Davos: 01/02/1997 – 12/31/2020   Hesse: 01/01/2014 – 12/31/2020   Fontainebleau: 03/11/2005 – 12/31/2022 | 31/03/2015 – 21/09/2023 | 01/01/2010 00:00 – 31/12/2022 23:00 | 01/2000 – 08/2023 | 01/1979 – 12/2018 |
| Characteristics | Standardized and filtered measurements from flux towers | Carbon model with 1km sub-grids and soil moisture assimilation | Dynamic global vegetation model forced by climate data (ERA5) | Machine learning based estimations, integrating satellite vegetation proxies, climate and flux tower measurements | |
| References | Pastorello et al. (2020); Warm Winter 2020 Team & ICOS Ecosystem Thematic Centre, 2022) | Jones et al. ( 2017); Kimball et al. (2022) | Smith et al., (2001, 2014); Wu (2023) | Zhuravlev et al. (2022) | Jung et al. (2019, 2020) |

**Table 1:** Summary of the datasets used in this study.





The two data-driven models use machine learning algorithms for upscaling and make use of observed $CO_2$ fluxes
from the FLUXNET network. The first data-driven model has been developed by the CarbonSpace company. It
makes use of (i) a Lagrangian particle dispersion model to account for the footprint of each tower flux site and (ii)
a gradient-boosted decision tree based non-linear regression (Chen, 2016) to derive one statistical model per land-
cover class. This approach follows that described in Zhuravlev et al. (2022), but with a revised regression
methodology and without use of meteorological variables. The Hesse flux tower site is not part of the 84 stations
in the FLUXNET2015 dataset used in the model input. A cross-validation is thus possible with Hesse and with
measures made after 2015 for the other sites (i.e. 7 years in Davos, 9 years in Fontainebleau). The current model
takes the aggregated Köppen–Geiger climate map at 1-km resolution (Beck et al., 2018) as a static predictive
variable, but does not yet include temporal climate variability. It provides monthly NEE only but at a very high
spatial resolution (few hectares) from 01-2000 to 08-2023. This allows to get as close as possible to the 3 flux
tower sites (around 1.8 ha centered on each tower) and their associated $CO_2$ flux measurement footprints, while
also distinguishing each non-instrumented plot (see section 2.1 for details on the area considered).

The second data-driven model comes from the FLUXCOM products (Tramontana et al., 2016; Jung et al., 2019,
2020) retrieved from the data portal of the Max Planck Institute for Biochemistry (https://www.bgc-jena.mpg.de).
The FLUXCOM products use eddy-covariance data from 224 flux-tower sites from the FLUXNET La Thuile
dataset (http://fluxnet.fluxdata. org/data/la-thuile-dataset/) and the CarboAfrica network (Valentini et al., 2014),
including Hesse data between 1997 and 2006 and Fontainebleau between 2005 and 2006. Cross-validations are
thus possible with most of our data from the Warm Winter 2020 release. The FLUXCOM products have been
shown to accurately estimate the mean annual and seasonal cycles of $CO_2$ fluxes (Tramontana et al., 2016; Jung
et al., 2020; He et al., 2022). Among the various forcing datasets available, we retained three of them, all forced
by hourly meteorological data from the ERA5 reanalysis (Hersbach et al., 2020) and providing global maps of
monthly NEE, GPP and RECO derived with a daytime partitioning on a 0.5° x 0.5° horizontal grid for the 1979-
2018 period. As for FLUXNET, the partitioning method does not significantly affect the $CO_2$ fluxes (Fig. A2).
The three datasets differ according to the algorithm used to build the statistical model: Random Forest (RF;
Breiman, 2001), Multivariate Adaptive Regression Splines (MARS; Friedman, 1991) and Artificial Neural
Networks (ANNs; Papale and Valentini, 2003). Unlike the CarbonSpace model, their coarse horizontal resolution
precludes the ability to account for individual forest stands. Despite these limitations, the three FLUXCOM
datasets allow to assess uncertainties induced by the statistical model used for upscaling $CO_2$ fluxes and to get
access to NEE and its two components.

The two process-based models are the Lund-Postdam-Jena General Ecosystem Simulator (LPJ-GUESS; Smith et
al., 2001, 2014) and the version 7 of the NASA Soil Moisture Active Passive Mission Level 4 Carbon (SMAP-
L4C; Jones et al., 2017; Kimball et al., 2022) models. The LPJ-GUESS is a dynamic global vegetation model
simulating the effects of environmental change in vegetation represented by plant functional types (PFTs), soil
hydrology and biogeochemistry (Smith et al., 2001). The model is widely used to study ecosystems, including $CO_2$
fluxes (Smith et al., 2001, 2014; Bayer et al., 2015; Lindeskog et al., 2021; Sathyanadh et al., 2021; Bergkvist et
al., 2023). The simulations used here were derived from Wu (2023) using version 4 of LPJ-GUESS in cohort mode
forced with hourly ERA5-land reanalysis (Muñoz-Sabater et al., 2021) and observed atmospheric $CO_2$



concentrations. The cohort mode means that woody plants of the same size and age are represented by a single
average individual. Each PFT is represented by multiple average individuals, and one PFT cohort is defined as the
average of several individuals. We retrieved hourly NEE, GPP and RECO on a 0.5° x 0.5° horizontal grid for the
2010-2022     period     from     the     ICOS     website     (https://meta.icos-
cp.eu/collections/NZNSUglRn0VeXmGDovuVY0ec). Like the FLUXCOM products, the horizontal resolution of
LPJ-GUESS outputs is too coarse to distinguish plots over the Mont Beuvray and Châtillonnais.

The SMAP (Soil Moisture Active Passive) Level 4 Carbon model product (SMAP-L4C) is produced operationally
by the NASA SMAP mission and can be considered as a reanalysis product since it uses the Goddard Earth
Observing System version 5 (GEOS-5) land model to assimilate SMAP L-band microwave observations and is
forced with observed land cover and vegetation from the Moderate Resolution Imaging Spectroradiometer
(MODIS) and Visible Infrared Imaging Radiometer Suite (VIIRS). The global processing is conducted on 1 km
sub-grids using spatially aggregated MODIS PFTs and VIIRS fPAR inputs, allowing to distinguish up to eight
individual PFTs within each 9 km x 9 km product grid cell. However, the model processing uses coarser spatial
resolution (9 km and 0.25 degree) daily inputs from the SMAP L4 soil moisture (L4_SM) and GMAO Forward
Processor (FP) surface meteorology. Among other variables, the SMAP-L4C outputs provide daily NEE and GPP
(RECO deduced from the difference between NEE and GPP), in a consistent global grid from March 2015 to
September 2023 for each PFT, including DBFs and ENFs (Jones et al., 2017; Kimball et al., 2022). The 1-km PFT
subclass distinction allows to differentiate ENF and DBF behavior over the Mont Beuvray plots. The L4C product
is derived using coupled photosynthetic light-use efficiency and soil organic matter decomposition models to
estimate daily NEE and it's component carbon fluxes; where, GPP is reduced from PFT-specific optimal rates for
unfavorable daily climate conditions including cold temperatures, low light levels, excessive atmospheric vapor
pressure deficits and low root zone (0-1m depth) soil moisture levels defined from SMAP L4_SM and GMAO FP
meteorology. Details of the model algorithms and the calibration, validation, and performance of the L4C version
7 product used in this study are given in the associated product quality assessment report (Endsley et al., 2023).

**2.3 Climate data**

Climate parameters are extracted from the version 2 of the operational chain Safran-ISBA-Modcou (SAFRAN-
SIM2; Soubeyroux et al., 2008). SAFRAN-SIM2 is an hydrometeorological reanalysis produced by Météo-France
at a 8 km spatial resolution from 1958 onwards. For each of the five sites, we extracted the nearest grid point for
2 m temperature (T in °C), soil water index of the first two meters (SWI in %), liquid, solid and total precipitation
(PRELIQ, PRENEI and PRE_SUM in mm), real and potential evapotranspiration (EVAP and ETP in mm) and 2
m relative humidity (HU in %). In addition, we calculated the air Vapor Pressure Deficit (in Pa), an integrative
metric accounting for both heat and water stress effects (Carrara et al., 2004; von Buttlar et al., 2018; Kong et al.,
2022; van der Woude et al., 2023). The VPD is defined as the difference between the amount of moisture that is
actually in the air and the amount of moisture that air could hold at saturation. The VPD is computed using the
Tetens formula (Monteith and Unsworth, 2007) following Eq. (1):
$$VPD = \left(1 - \frac{HU}{100}\right) * saturation\ vapor\ pressure = \left(1 - \frac{HU}{100}\right)\left(610.78 * \exp\left(\frac{T}{T+237.3} * 17.2694\right)\right) \quad (1)$$



Preliminary analyses show that the SAFRAN-SIM2 reanalysis accurately captures the temporal variability and
magnitude of 2 m temperature and precipitation compared to observations provided by the three FLUXNET sites
(Fig. A3), despite biased solid precipitation in Davos. For this reason and for conciseness, we consider only
SAFRAN-SIM2 regardless of the site and $CO_2$ flux product.

**2.4 Methodology**

For the gridded datasets (SAFRAN-SIM2, FLUXCOM, LPJ-GUESS and SMAP-L4C), we extracted the nearest
grid point to the flux tower sites and to the center of Mont Beuvray and Châtillonnais plots. Since all datasets have
different temporal resolution and units (Table 1), they all have been converted to $tCO_2$ ha$^{-1}$ month$^{-1}$ and aggregated
at the monthly timescale. From these monthly values, we computed the mean annual cycle by averaging all years
available in each dataset, as well as its interannual variability defined as the standard deviation of monthly values.
The annual budget was calculated as the sum of the monthly values, only for complete years (i.e. when no monthly
value is missing). Fontainebleau is the only site presenting gaps in the observed time series (in 2005, 2014 and
2017) due to too low-quality control values. The mean annual budget is then computed together with its interannual
variability following the same procedure described above.

The model skill in capturing observed $CO_2$ flux temporal variability at the monthly and annual timescales is
assessed over overlapping periods between each model end each observation. Magnitude and co-variability errors
are assessed in terms of bias and Bravais-Pearson correlation coefficient (R) or coefficient of determination ($R^2$),
respectively. The evaluation is done considering raw monthly values to focus on the annual cycle, as well as
monthly anomalies (i.e., removal of the mean annual cycle) and raw annual values to focus on interannual
variability at the monthly and annual timescales, respectively.

The R and $R^2$ metrics are also used to assess the relationship between climate variables and $CO_2$ fluxes at the
monthly (raw and anomalous values) and annual (raw values) timescales. In addition, a composite approach is
performed to examine monthly climate anomalies associated with large negative and positive monthly anomalies
in $CO_2$ fluxes (NEE, GPP and RECO). Large negative/positive $CO_2$ flux anomalies are defined as standardized
anomalies (mean=0, standard deviation=1) below/above -0.5/+0.5. Tests with stricter threshold values (e.g., -1/+1)
lead to similar results but limit the size of the samples. The difference between the two groups is tested for
significance based on the non-parametric Mann-Whitney *U* test (McKnight and Najab, 2010).

**3 Results**

**3.1 Monthly timescale**

**3.1.1 Mean annual cycle and interannual variability in climate and $CO_2$ fluxes**



Figure 2 shows the mean annual cycle and interannual variability of T and main surface water cycle parameters
associated with each site. All sites follow similar patterns of T, ETP, EVAP and VPD with the greatest values in
summer and the lowest in winter. The annual cycle in SWI is reversed, with drier soils in late summer and wetter
soils in winter. The total precipitation is evenly distributed throughout the year for sites in plain (Fontainebleau
and Hesse and Châtillonnais), in contrast with mountainous sites (Davos and Mont Beuvray) where precipitation
amounts are larger during winter than summer. The interannual variability (shadings on Fig. 2) is particularly
pronounced all year long for PRE_SUM and from spring to fall for VPD, highlighting strong year-to-year
fluctuations of the water cycle.

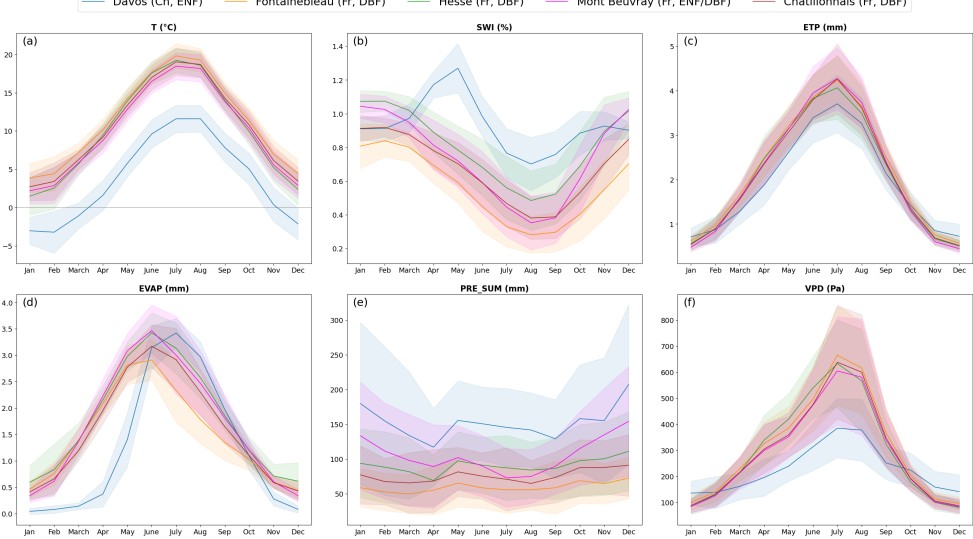

**Figure 2:** Mean annual cycle and interannual variability in monthly (a) 2 m air temperature (T), (b) soil moisture
of the first two meters (SWI), (c) potential evapotranspiration (ETP), (d) real evapotranspiration (ETR), (e) total
precipitation (liquid + solid: PRE_SUM) and (f) vapor pressure deficit (VPD) for each study site (colors, see insert)
for the 1990–2023 period. Climate conditions in each site are extracted from the nearest grid point of the 8 km x
8 km SAFRAN-SIM2 reanalysis. Bold lines show the mean annual cycle. Shadings show interannual variability
computed as the standard deviation of each month of the period.

Due to its much higher elevation, Davos depicts different climate conditions than the other sites with (i) lower T
by up to ~10 °C all year long, (ii) wetter soils, especially in spring due to mild temperature, low evaporation and
snow melting (not shown), (iii) larger precipitation amounts all year long with snowfall from October to April (not
shown) and (iv) delayed EVAP peak in late summer. While this site is not an analogue of the Mont Beuvray ENF
site, it remains the most representative one available in the FLUXNET network.

The mean annual cycle of monthly NEE is marked in all study sites (Fig. 3). Temperate forest ecosystems release
$CO_2$ during winter and sequester $CO_2$ during summer, with higher values in summer than in winter. While this
overall cycle prevails all years, the sign of the NEE can be reversed from one year to another in spring and fall in





almost all products and sites, indicating that during these seasons the forest ecosystems can be either a $CO_2$ source
or a $CO_2$ sink. Seasonal contrasts are stronger in DBF than ENF sites, consistent with the DBF leafing seasonality
and with previous studies hypothesizing a buffer effect in ENF soils (e.g., Welp et al., 2007; Zheng et al., 2021).
The magnitude of interannual variability seems also to be influenced by forest stand characteristics (e.g., in Mont
Beuvray in the SMAP-L4C and CarbonSpace models), with e.g. a variability 40% higher for DBFs than ENFs
simulated by the CarbonSpace model in Mont Beuvray in July. Despite high coupling between GPP and RECO,
the NEE mean annual cycle is mostly driven by GPP in summer and by RECO in winter regardless of the sites, as
illustrated for one DBF site in Fig. 4.

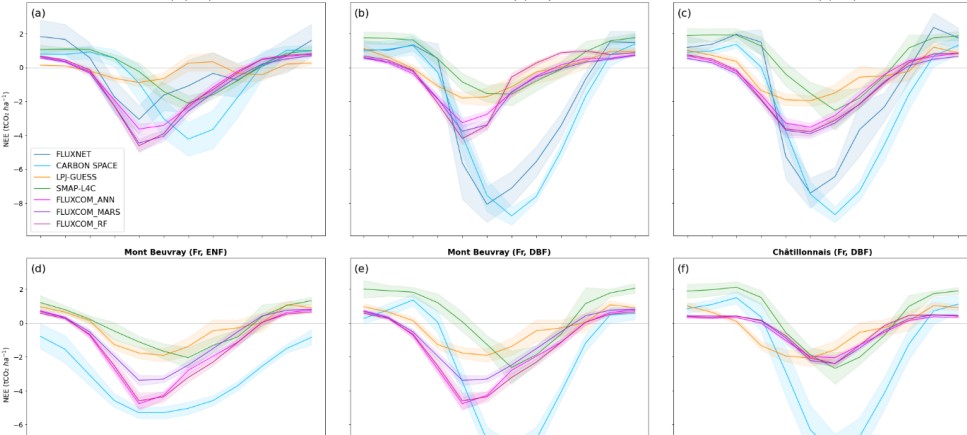

**Figure 3:** Same as Fig. 2 but for monthly NEE in the (a-c) three FLUXNET sites and (d-f) three non-instrumented
sites located in Burgundy as measured by eddy-covariance (FLUXNET) and simulated by data-driven
(CarbonSpace and FLUXCOM) and process-based (LPJ-GUESS and SMAP-L4C) models. The longest available
period for each site and dataset is retained. See Table 1 for details. For LPJ-GUESS, SMAP-L4C and FLUXCOM
models, the nearest grid point from each site is shown. Results from LPJ-GUESS and FLUXCOM are similar in
panels (d-e) due to coarse horizontal resolution (0.5° x 0.5°) and no distinction between forest stands (DBF or
ENF).





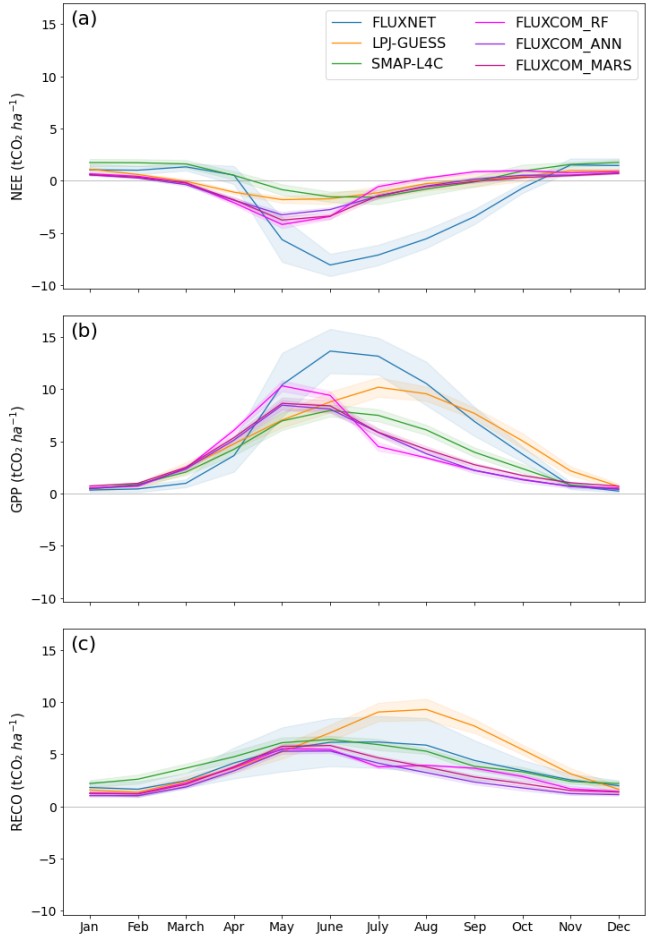

**Figure 4:** Same as Fig. 3 but for monthly (a) NEE, (b) GPP and (c) RECO in the Fontainebleau DBF site according to FLUXNET observations, FLUXCOM data-driven and LPJ-GUESS and SMAP-L4C process-based models. The CarbonSpace model is not shown since GPP and RECO are not available for this model.

The model skill in capturing monthly NEE depends on the metrics, sites and products (Fig. 5). Overall, the annual cycle in NEE is better represented in the two DBF sites, located in the plain, than the ENF site, Davos, located in the Alps Mountain area (Fig. 5a-c). The reverse holds in terms of magnitude (Fig. 5d-f). In Davos, all models reasonably capture the magnitude of monthly NEE, with inter-quartile in the -2 and +2 $tCO_2$ ha$^{-1}$ range, while the FLUXCOMs are the only models to reasonably capture the NEE annual cycle ($R^2$ inter-quartile in the 0.6-0.7 range). In the two DBF sites, the CarbonSpace data-driven model clearly provides the best scores for both metrics ($R^2$ in the 0.8-0.9 range and bias in the $0 - -2$ $tCO_2$ ha$^{-1}$ range), suggesting an added value of very high spatial resolution upscaling for representing $CO_2$ flux annual cycle and magnitude. The remaining models strongly underestimate the $CO_2$ uptake during summer and $CO_2$ release during winter, with 25% of the NEE values




associated with biases exceeding 3-4 $tCO_2$ $ha^{-1}$ in both DBF sites. Interestingly, the model deficiencies in capturing
(i) the annual cycle in NEE is mainly linked to poorly resolved temporal variability of RECO (Fig. A4) and (ii)
the biased NEE magnitude is mainly linked to underestimated $CO_2$ uptake by photosynthesis (Fig. A5).

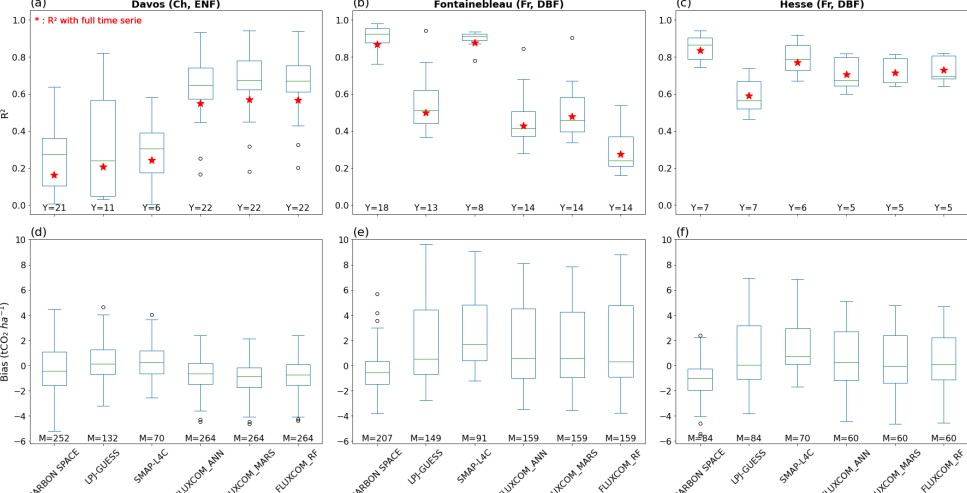

**Figure 5:** Skill of data-driven and process-based models in capturing the (a-c) annual cycle and (d-f) magnitude
in monthly NEE for each FLUXNET site. (a-c) The quality of the annual cycle is assessed through the coefficient
of determination ($R^2$) between simulated and observed monthly NEE (12 values) computed for each overlapping
year (labeled on panels a-c). (d-f) The magnitude errors are computed as the difference (i.e., bias) between
simulated and observed NEE for each month of the overlapping period (labeled on panels d-f). The boxes have
lines at the lower quartile, median, and upper quartile values. The whiskers are lines extending from each end of
the boxes to show the extent of the range of the data within 1.5 by inter-quartile range from the upper and lower
quartiles. Circles are outliers and red crosses in panels (a-c) are $R^2$ computed considering the full monthly
timeseries at once (e.g., 252 months for the first from left boxplot in panel a).


The annual cycle and magnitude of NEE simulated in the ENF and the two DBF non-instrumented plots (Fig. 3d-
f) closely resemble those simulated in the instrumented sites (Fig. 3a-c). The two main exceptions concern all-
year-long $CO_2$ sequestration simulated by the CarbonSpace model in the Mont Beuvray ENF site, and much
weaker $CO_2$ sequestration peak simulated by the FLUXCOM models in the Châtillonnais DBF site. Based on these
results, the CarbonSpace data-driven model appears to be the best compromise to capture the annual cycle and
magnitude of NEE associated with both ENFs and DBFs. The results also demonstrate that process-based and
data-driven models have their own strengths and weaknesses and are thus complementary.

**3.1.2 Climate – CO₂ flux relationship**

The $CO_2$ flux – climate relationship is assessed considering both raw monthly values to account for the annual
cycle and monthly anomalies (i.e., mean annual cycle removed) to focus on interannual variability.





The synchronous relationship between the annual cycle of $CO_2$ fluxes and that of various climate parameters is
assessed through a correlation analysis of raw monthly time series (Fig. 6). Regardless of the sites and models, T,
VPD, ETP and EVAP correlate the most with NEE. For these parameters, the correlation coefficients are negative
meaning that $CO_2$ uptake increases with higher T, VPD and evapotranspiration. The weakest correlations are found
for PRE_SUM and, to a lesser extent, SWI. Although we can hypothesize that climate variables are interdependent,
a variance inflation factor (VIF) calculation highlighted the existence of multicollinearity only between ETP and
VPD (VIF>5 in almost all sites and models).

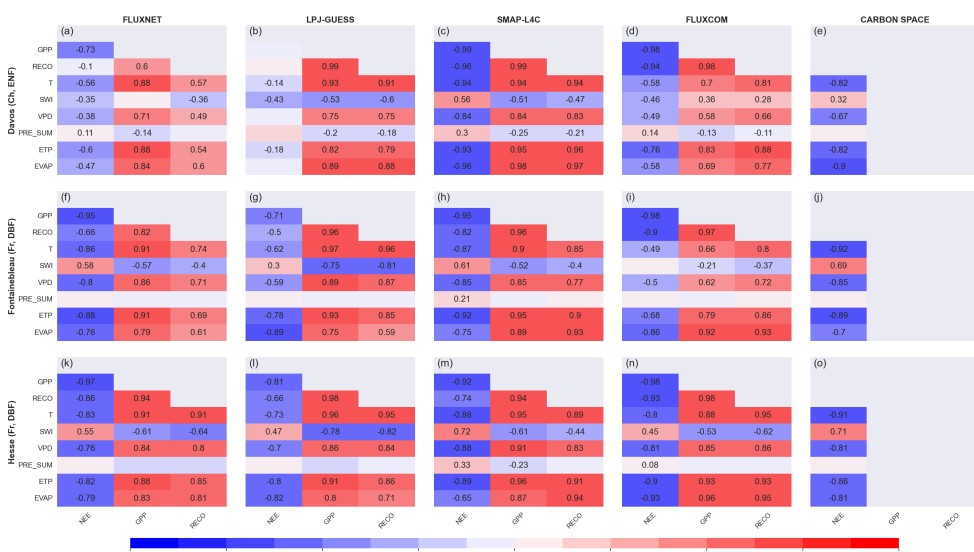


**Figure 6:** Bravais Pearson correlation coefficient values (shadings) between the three $CO_2$ flux variables from
each model and site, and the associated SAFRAN-SIM2 climate parameters associated with each site location.
Climate parameters include 2 m air temperature (T in °C), soil water index (SWI in %), vapor pressure deficit
(VPD in Pa), total precipitation (PRE_SUM in mm), potential and real evapotranspiration (ETP and EVAP in
mm). (a-e) Davos, (f-j) Fontainebleau and (k-o) Hesse FLUXNET sites. Correlations are computed considering all
months and all years for overlapping periods between climate and $CO_2$ flux datasets. See Table 1 for details. The
FLUXCOM multi-model mean is shown in panels d,i,n for conciseness since the three FLUXCOM models provide
similar results. GPP and RECO are not shown in panels e,j,o because they are not provided by the CarbonSpace
data-driven model. Only correlation values significant at the 90% confidence level are written.


Three main results emerge when comparing the influence of climate on the annual cycle in NEE, GPP and RECO.
First, the climate influence is most of the time greater on GPP than RECO, particularly in the FLUXNET
observations, meaning that photosynthesis is more affected by climate conditions than are respiration processes.
Second, the climate influence can be strong on both GPP and RECO but weak on NEE (Fig. 6). This is for instance
the case in Davos where the correlation coefficient between T and LPJ-GUESS-simulated $CO_2$ fluxes exceeds 0.9
for both GPP and RECO while remains weak and barely significant for NEE. Last, the climate influence on the



annual cycle in simulated $CO_2$ fluxes highly depends on the model capability to capture the $CO_2$ flux annual cycle.
From this point of view, the CarbonSpace data-driven and SMAP-L4C process-based models often depict stronger
$CO_2$ fluxes – climate relationships than the remaining models, consistent with more accurate simulation of the $CO_2$
flux annual cycle (Fig. 3).

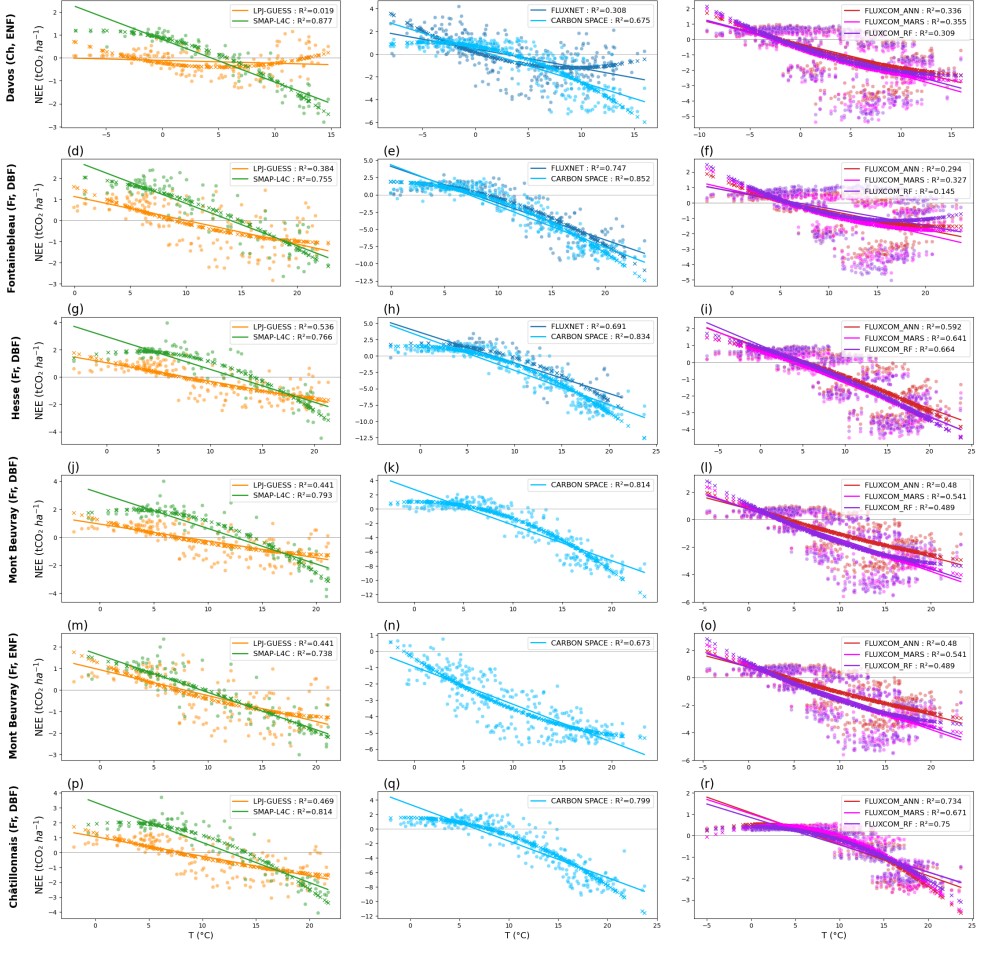

**Figure 7:** Simple linear and $2^{nd}$ order polynomial regressions between monthly 2 m temperature from SAFRAN-SIM2 (x-axis) and NEE (y-axis) from all datasets and sites. (a,d,g,j,m,p) LPJ-GUESS and SMAP-L4C process-based models. (b,e,h,k,n,q) FLUXNET observations and CarbonSpace data-driven model. (c,f,i,l,o,r) FLUXCOM data-driven models. The coefficient of determination ($R^2$) in the insert is derived from the linear regression.



A particular attention is given on the relationship between T and NEE (Fig. 7) to further discuss uncertainties
induced by the products and dependencies to forest stand conditions. The relationship is systematically weaker (i)
in Davos than in other sites regardless of the product and (ii) in the two coarse resolution models (LPJ-GUESS
and FLUXCOM) regardless of the site. The relationship is linear-like in ENF sites regardless of the dataset and





the CO$_2$ flux. This contrasts with DBF sites where the observed T – NEE relationship is polynomial with an evident
threshold effect. Below 10°C, the NEE turns positive, indicating a net ecosystem carbon loss, and stabilizes in the
0–2 tCO$_2$ ha$^{-1}$ range. This threshold effect corresponds to the low biological activity of DBFs under cold conditions,
hence weak to no CO$_2$ uptake by photosynthesis. This threshold effect results thus from GPP only, as reflected by
the flattening of the GPP curve at low T (Fig. A6) unlike the linear RECO – T relationship (Fig. A7). Among the
models, the SMAP-L4C and CarbonSpace are the only models to capture the observed threshold, highlighting the
usefulness of distinguishing the PFT in the model outputs.

The above analyses (Figs. 6-7) depict significant relationships between climate and CO$_2$ fluxes when accounting
for the annual cycle. Once the latter is removed (see section 1.4 for details), most of the correlation values are
higher for process-based than data-driven models, remain of the same sign but are of weaker magnitude for GPP
and RECO and are almost negligible for NEE (Fig. 8). Positive anomalies in GPP and RECO are associated with
positive anomalies in T, VPD and ETP and negative anomalies in soil moisture (SWI) and, to a lesser extent,
precipitation in most sites and products. The reverse holds true for negative GPP and RECO anomalies. The similar
response of GPP and RECO to the climate anomalies induces a compensation effect on the residual NEE carbon
flux, resulting in weak NEE anomalies. The sign of NEE anomalies is uncertain among the sites and the products
and depends on which of the two components is associated with the largest anomalies. While the models tend to
exaggerate the observed relationship between CO$_2$ flux and climate anomalies, especially the process-based
models, the overall picture is satisfactorily captured.

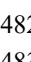

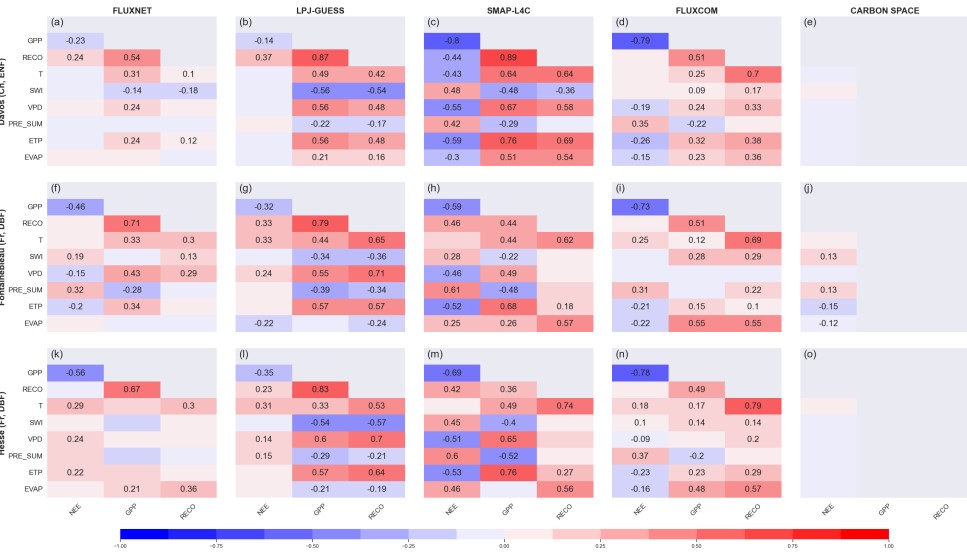


**Figure 8:** Same as Fig. 6 but after removing the mean annual cycle.


In addition to the correlation analysis, for which few disagreements can lead to poor correlation values, we now
investigate climate anomalies associated with the largest anomalies in monthly CO$_2$ fluxes (see Section 2.4 for



details). Results are shown for the Fontainebleau DBF site only for conciseness in Figs. 9-11. The main results can
be summarized as follows. Two main climate parameters, PRE_SUM and T, appear to significantly influence the
largest $CO_2$ flux monthly anomalies in almost all datasets. Wet anomalies significantly favor anomalies that are
positive for RECO and negative for GPP, hence less $CO_2$ uptake (Figs. 9c, 10c and 11c), and vice versa for dry
anomalies. Warm anomalies are associated with large positive NEE anomalies (i.e., strong $CO_2$ emissions or weak
$CO_2$ uptake) and vice versa for cold anomalies (Fig. 9a). In turn, the DBF ecosystem sequesters less (more) $CO_2$
during anomalously warm (cold) conditions. This result is more clearly driven by RECO (Fig. 10a) than GPP (Fig.
11a), consistent with an exponential response of respiration to T (van't Hoff, 1898). Some relationships are
opposite in sign between the process-based models and the data-driven models, highlighting strong uncertainties
induced by the approach. For instance, anomalously dry soil is associated with RECO and GPP anomalies that are
positive in the process-based models, especially LPJ-GUESS, while negative in the FLUXCOM data-driven
models (Figs. 10b and 11b).





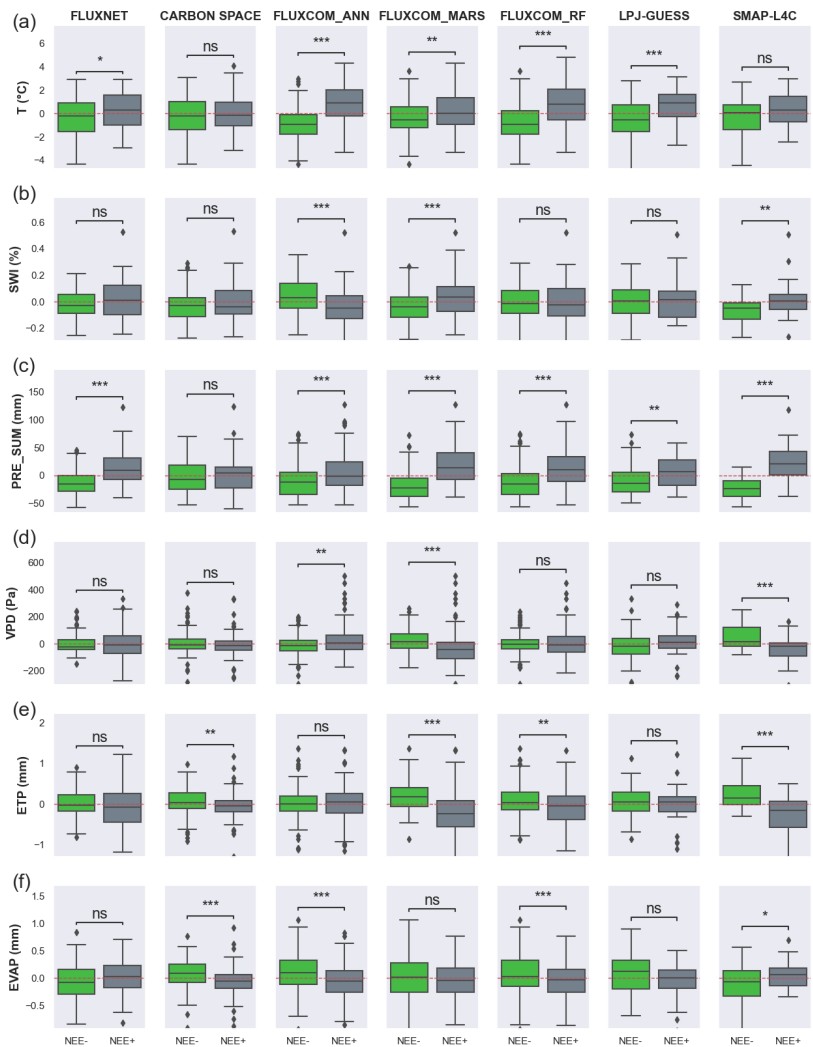

**Figure 9:** Monthly climate anomalies associated with large negative and positive anomalies in monthly NEE (NEE- and NEE+, respectively) for each dataset in the Fontainebleau DBF site. (a) 2m temperature (°C). (b) Soil water index (%). (c) Total precipitation (mm/month). (d) Vapor pressure deficit (Pa). (e) Potential evapotranspiration (mm/month). (f) Real evapotranspiration (mm/month). NEE- (NEE+) anomalies are defined as standardized anomalies (mean=0; standard deviation=1) below -0.5 (above 0.5). The boxes have lines at the lower quartile, median, and upper quartile values. The whiskers are lines extending from each end of the boxes to show the extent of the range of the data within 1.5 by inter-quartile range from the upper and lower quartiles. Circles are outliers. The symbol ns indicates no statistically significant difference in climate anomalies between NEE- and NEE+ according to a Mann-Whitney $U$ test. The symbols *, ** and *** correspond to significant differences at the 90, 95 and 99% confidence level according to the same test.





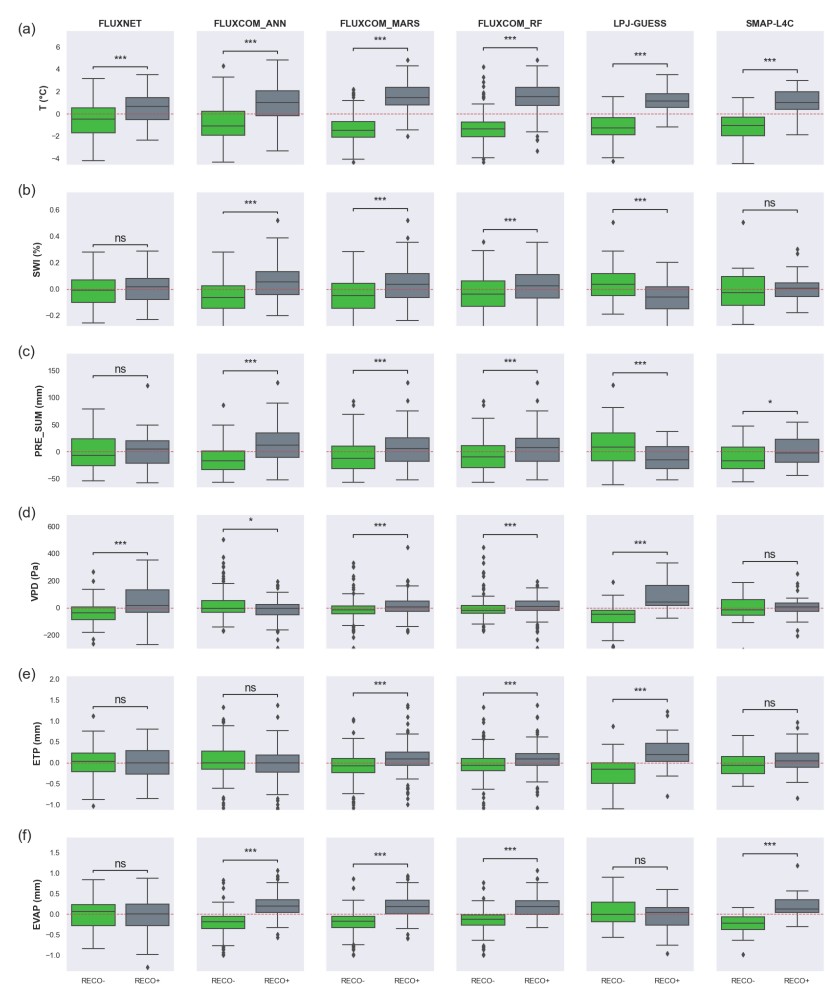

**Figure 10:** Same as Fig. 9 but for RECO.





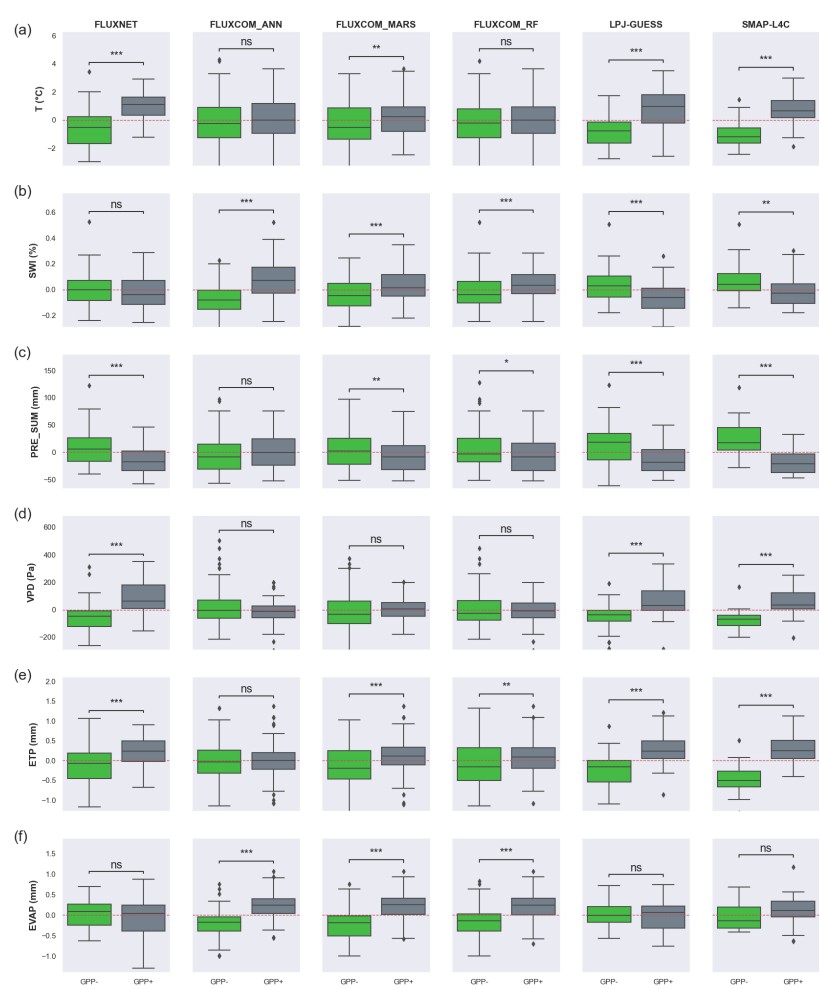

**Figure 11:** Same as Fig. 9 but for GPP.

This section demonstrates that the annual cycle of monthly $CO_2$ fluxes is sharply driven by climate, while its interannual variability is not a simple response to climate anomalies. Most of the models accurately capture the observed annual cycle and its relationship with climate. However, the interannual variability of monthly $CO_2$ flux anomalies are not necessarily phased between observations and models (i.e., weak co-variability), especially for NEE, and the models tend to exaggerate the impact of climate on $CO_2$ flux interannual variability.



### 3.2 Annual timescale

#### 3.2.1 Mean annual budget and interannual variability

In the observations, the mean annual $CO_2$ sequestration of the ecosystems is, on average, far weaker in the ENF site (-2.24 $tCO_2$ ha$^{-1}$ year$^{-1}$) than the DBF sites (-24.63 $tCO_2$ ha$^{-1}$ year$^{-1}$ in Fontainebleau and -15.14 $tCO_2$ ha$^{-1}$ year$^{-1}$ in Hesse), while the reverse holds in terms of interannual variability (±6.92 against 3.92 and 3.60 $tCO_2$ ha$^{-1}$ year$^{-1}$, respectively) (Fig. 12a). The large contrast between DBF and ENF sites may not be transposable since climate in Davos is atypical, with much colder and wetter mean conditions and larger year-to-year variability compared to the remaining sites (Fig. 2). Only the two process-based models (LPJ-GUESS and SMAP-L4C) provide mean annual $CO_2$ uptake values comparable to observations in Davos. However, this might be a coincidence since these models strongly underestimate the NEE annual budget in the other sites due to underestimated GPP (Fig. 12b) and overestimated RECO (Fig. 12c). In particular, the NEE annual budget is positive in all DBF sites in the SMAP-L4C owing to a too short duration of the uptake season simulated by this model (Fig. 3), resulting in annual GPP bias of e.g. -21.77 $tCO_2$ ha$^{-1}$ year$^{-1}$ in Fontainebleau (Fig. 12b).

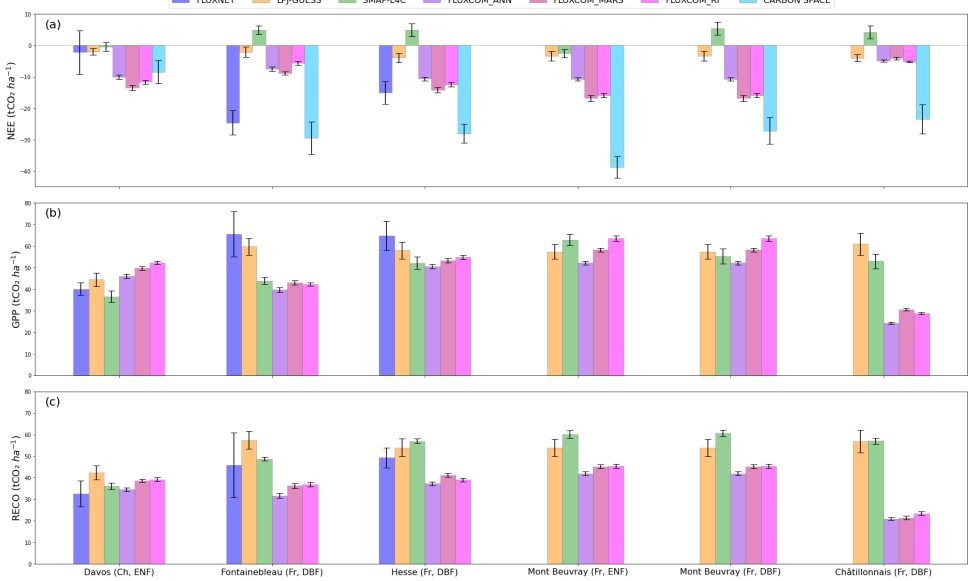

**Figure 12:** Mean annual budget (bars) and interannual variability (whiskers) in (a) NEE, (b) GPP and (c) RECO for each site and dataset. Note that the results are similar in the Mont Beuvray ENF and DBF sites for the LPJ-GUESS and FLUXCOM models due to their coarse resolution and no distinction of the forest stand in the outputs.

Except in Davos, the two data-driven models perform reasonably well to capture the observed magnitude of NEE annual budget, despite $CO_2$ uptake in DBF sites is underestimated by 15 to 20 $tCO_2$ ha$^{-1}$ year$^{-1}$ in Fontainebleau by the FLUXCOM models and overestimated by 19.99 $tCO_2$ ha$^{-1}$ year$^{-1}$ in Hesse by the CarbonSpace model. Over Mont Beuvray, the annual $CO_2$ uptake is greater in the ENF than the DBF site for the models distinguishing the



PFTs in their outputs (i.e., CarbonSpace and SMAP-L4C). This might be explained by a basal area reaching 49 $m^2$
$ha^{-1}$ in average in the ENF plot against 32 $m^2$ $ha^{-1}$ in the DBF one (for a volume of 656 $m^3$ $ha^{-1}$ and 404 $m^3$ $ha^{-1}$;
data based on forest inventory), inducing a weaker photosynthetic activity in the DBF (Fig. 12b). This hierarchy
cannot be captured by the FLUXCOM and LPJ-GUESS models by construction since the two Mont Beuvray plots
are part of the same grid point and ENF and DBF are not distinguished.

At the interannual timescale, the forest ecosystem systematically acts as a $CO_2$ sink (Fig. 12), except (i) in Davos
for the observations and the SMAP-L4C model and (ii) in all DBF sites for the SMAP-L4C model where NEE is
always positive (i.e., $CO_2$ release), as already discussed. The magnitude of interannual variability in annual NEE
is the largest and the closest to the observed one for the CarbonSpace model (±3.92 and 5.16 $tCO_2$ $ha^{-1}$ $year^{-1}$ in
Fontainebleau), and the lowest and farthest from the observed one for the FLUXCOM models (±0.55 $tCO_2$ $ha^{-1}$
$year^{-1}$ in Fontainebleau with FLUXCOM_MARS), a statement also prevailing for GPP and RECO.

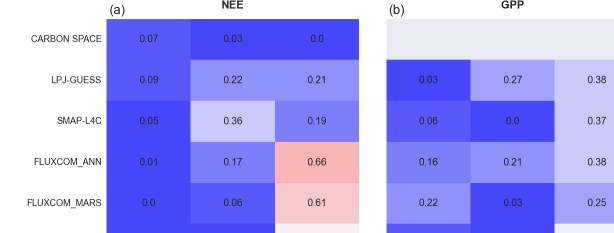
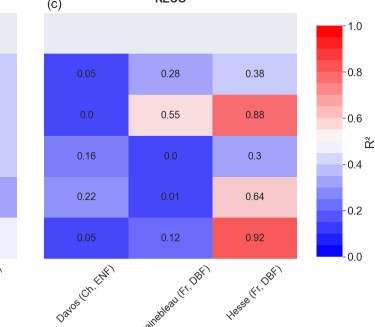

**Figure 13:** Coefficient of determination (numbers and shadings) between modelled and measured fluxes in the
FLUXNET sites at the annual timescale.


Beyond the magnitude of interannual variability, a critical question concerns the model capability to capture the
observed year-to-year fluctuations of annual $CO_2$ fluxes (Fig. 13). No model succeeds at capturing the observed
NEE, GPP and RECO interannual variability in Davos, suggesting deficiencies of state-of-the-art models in
simulating $CO_2$ flux interannual variability in mountainous regions. At least for SMAP-L4C, this may be due to
the inability of the coarse (0.25 degree resolution) GEOS FP daily meteorology and (9-km grid) SMAP L4 soil
moisture to capture the larger spatial heterogeneity in local climate conditions imposed from the complex mountain
terrain at this site. For the remaining sites, the models tend to better perform with GPP and RECO than NEE. The
CarbonSpace model fails at capturing the observed interannual variability in annual NEE ($R^2 \leq 0.08$). Despite biased
annual mean conditions (Fig. 12), the SMAP-L4C performs reasonably well, with $R^2$ of 0.36 and 0.55 for NEE
and RECO in Fontainebleau and 0.88 for RECO in Hesse (Fig. 13). This model is the only one to be forced by
satellite observation informed soil moisture, suggesting this parameter is valuable for simulating realistic year-to-
year fluctuations of annual $CO_2$ fluxes. The FLUXCOM models also capture correctly the interannual variability
in Hesse for all $CO_2$ fluxes and perform better (with still low $R^2$) in Davos for GPP and RECO than the other





models. Importantly, this skill is not "forced" by construction since the Davos site is not used to train the
FLUXCOM models, and there is no or few (2) overlapping years between the period used to train the models and
that analyzed in our study for Hesse and Fontainebleau. The LPJ-GUESS provides intermediate scores, with $R^2$ in
the $0.3 - 0.4$ range. These results indicate that (i) models accounting for climate variability better capture
interannual variability in $CO_2$ fluxes and (ii) the simulated interannual variability is closer to observations at the
annual than monthly timescale.

**3.2.2 Relationship with climate**

Overall, the interannual co-variability between $CO_2$ flux and climate is qualitatively similar at the annual (Fig. 14)
than the monthly (Fig. 8) timescale, with correlation values of the same sign. The main difference concerns less
significant correlation values at the annual than monthly timescale for most variables, probably due to a sample
size effect since the number of years under study is limited. The only exception concerns a stronger precipitation
– $CO_2$ flux relationship at the annual than monthly timescale for the two process-based models, especially for GPP
and RECO with decreased $CO_2$ fluxes during wet years.

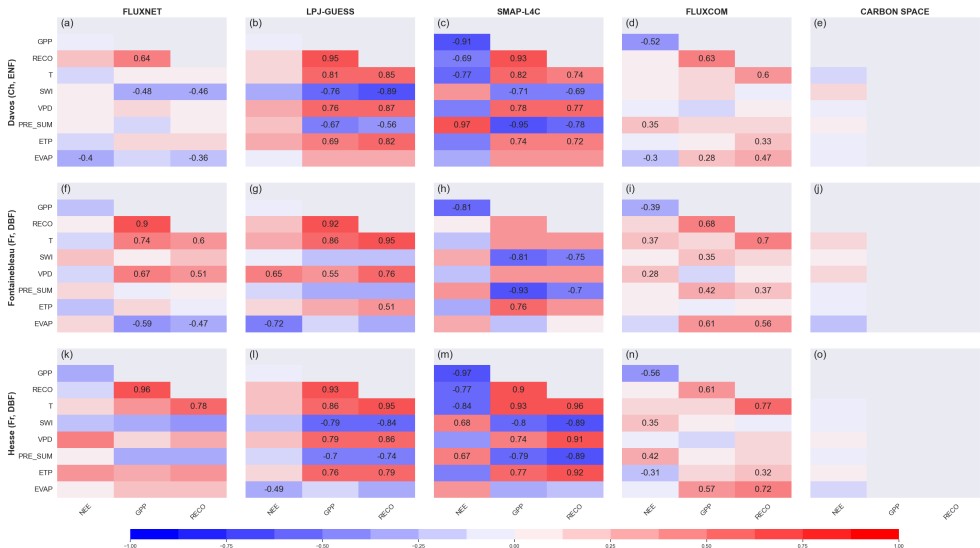

**Figure 14:** Same as Fig. 6 but for raw annual values.


The "long-term" evolution in the NEE annual budget does not depict any trend but is characterized by strong year-
to-year fluctuations, except for the FLUXCOM models depicting surprisingly flat variability (Fig. 15). A modest
increase is looming in the very last years of FLUXNET data, especially in DBF plots (Fig. 15b-c) but, as for all
other models, the temporal coverage seems too short and the interannual variability too strong to settle any
conclusion. Our result contrasts with recent literature pointing towards reduced $CO_2$ uptake in Europe (Smith et
al., 2020; Thompson et al., 2020; Chuine et al., 2023; van der Woude et al., 2023). Possible reasons involve the



limited number of sites under study, the fact that eddy-covariance flux tower measurements may be located in
healthy forest ecosystems and potential compensation effects between GPP trends and RECO trends. The latter
point is critical, at least in the FLUXNET observations and LPJ-GUESS simulations (Figs. A8-A9). To further test
this hypothesis, Figure 16 shows the temporal evolution of annual anomalies in observed $CO_2$ fluxes and climate
parameters in the Fontainebleau DBF site. While annual NEE anomalies do not depict any trend (Fig. 16a), GPP
and RECO anomalies are most frequently negative before 2014 and positive afterwards (Fig. 16b). The time series
is too short to conclude whether such an evolution is reminiscent of a trend or a decadal-like variability. However,
this pattern is consistent with the evolution of annual climate anomalies depicting drier conditions, larger potential
evapotranspiration and colder temperature before 2014 than afterwards (Fig. 16c-e).

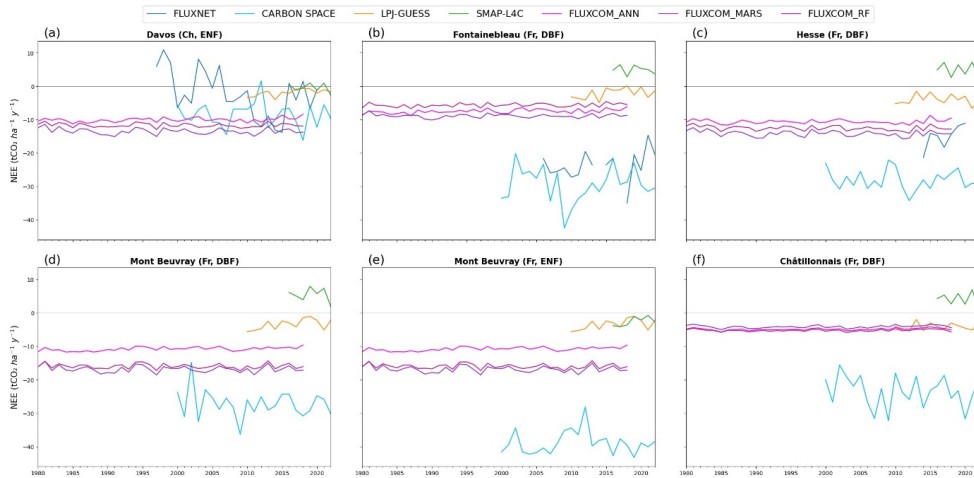


**Figure 15:** Annual NEE for the longest available period for each site and dataset. Gaps are due to not complete
years (e.g. FLUXNET data in Fontainebleau in panel b).



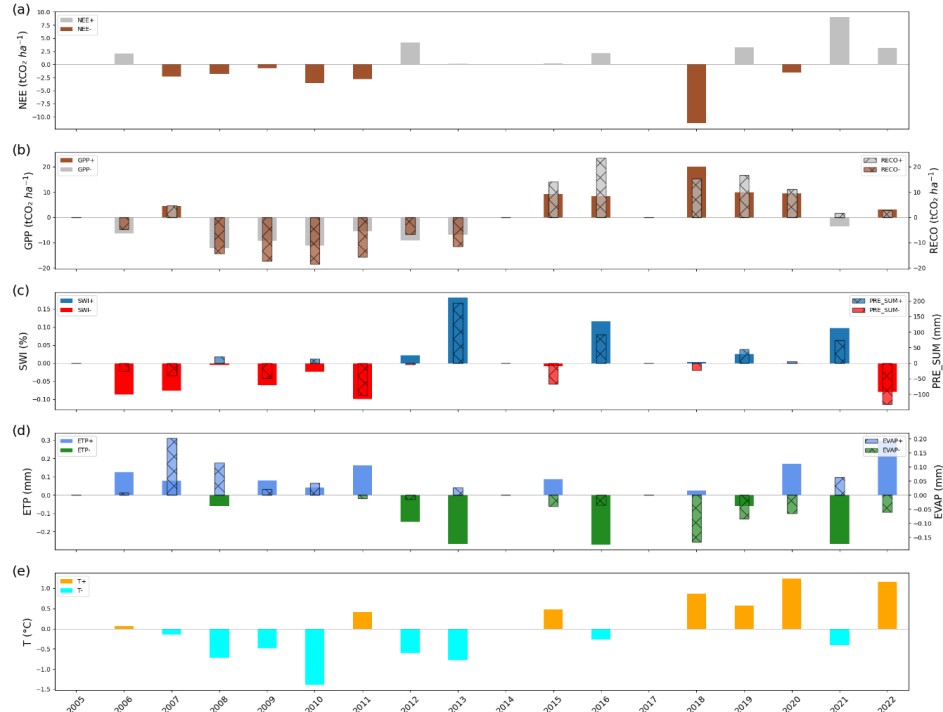



**Figure 16:** Evolution of annual anomalies in observed $CO_2$ budget and SAFRAN-SIM2 climate in Fontainebleau
between 2005 and 2022. (a) NEE. (b) GPP and RECO. (c) Soil moisture and total precipitation. (d) Real and
potential evapotranspiration. (e) 2 m temperature. Anomalies are computed as the difference between each year
and the 2005-2022 averaged conditions. The years 2005, 2014 and 2017 have not been accounted for because of
missing $CO_2$ flux data.

643

644

**4 Discussion**

646

This study aims at evaluating process-based and data-driven models in capturing $CO_2$ flux temporal dynamics of
temperate forest ecosystems and their relationships with climate. Such an evaluation is required to question the
extent to which these models may provide relevant information for monitoring $CO_2$ temporal dynamics and
understanding their drivers in temperate forests where no $CO_2$ measure is available.

651

First, we show that the model skill depends on the target. On the one hand, the magnitude and pattern of the annual
cycle, annual budget and the range of interannual variability (i.e., standard deviation of monthly or annual values)
are better captured by the CarbonSpace data-driven model than the remaining models. This added value was
expected in e.g. Fontainebleau, since this site is included in the pool of flux tower measurements used for the
model calibration, but not in Hesse since this site is not included. Furthermore, the CarbonSpace clearly
outperforms the other data-driven models tested (FLUXCOM models set with different AI algorithms), which are



also calibrated with flux tower measurements. This suggests that the accurate skill of the CarbonSpace model relies
also on the inclusion of high resolution multi-spectral satellite data allowing to assess $CO_2$ dynamics at the
hectometric scale and to distinguish different PFTs. On the other hand, the co-variability between observations
and models, and between $CO_2$ fluxes and climate depends on whether the focus is on the annual cycle or on the
interannual variability. When focusing on the annual cycle (Figs. 4-7), the co-variability is the largest for models
capturing the observed annual cycle in $CO_2$ fluxes, with CarbonSpace providing the best scores. When focusing
on the interannual variability of monthly anomalies and annual budgets (Figs. 8 and 14), models forced by dynamic
climate data (LPJ-GUESS, SMAP-L4C and FLUXCOM) clearly outperform the CarbonSpace model, which is
forced by static climate data. In particular, the SMAP-L4C provides satisfactory results, suggesting that soil
moisture is a key parameter for monitoring the interannual variability of $CO_2$ fluxes.

Second, we show that the $CO_2$ flux – climate relationship is stronger for GPP and RECO than NEE and that the
sign of the relationship between GPP/RECO and climate is relatively similar among the sites and products both
along the annual cycle and from year-to-year (monthly anomalies and raw annual budgets). Both RECO and GPP
increase when 2 m temperature, vapor pressure deficit and evapotranspiration increase and when precipitation and
soil moisture decrease, in line with the literature (Haszpra et al., 2005; Tang et al., 2013; Kong et al., 2022; Li et
al., 2023; Sharma et al., 2022). The NEE – climate relationship is more complex. Along the annual cycle, NEE is
mainly driven by RECO during winter and GPP during summer in all datasets. From year-to-year, the magnitude
of NEE anomalies is weak in most cases and their sign depends on the magnitude of the response of GPP and
RECO. The latter point induces site dependencies and disagreements between observations and models and
between models. From this point of view, models providing the three $CO_2$ fluxes (i.e., NEE, GPP and RECO)
allow for a better understanding on $CO_2$ exchanges between the atmosphere and forest ecosystems. In addition,
our study focuses on the synchronous relationship between $CO_2$ fluxes and individual climate parameters.
Considering lead-lag relationships as well as the influence of combined climate parameters would be the next step
to account for the long term effect of droughts and heatwaves on forest ecosystems (Ciais et al., 2005; von Buttlar
et al., 2018). Similarly, the number of study sites was too limited to account for the influence of variable soil
properties (Kurbatova et al., 2008; Besnard et al., 2018; Curtis and Gough, 2018; Martinez del Castillo et al.,
2022), forest management practices (Carrara et al., 2003; Scott et al., 2004; Saunders et al., 2012) and stand age
(Kurbatova et al., 2008; Besnard et al., 2018; Chuine et al., 2023).

Third, distinguishing forest stands is critical for a fine scale assessment of $CO_2$ temporal variability (Carrara et al.,
2003, 2004; Welp et al., 2007; von Buttlar et al., 2018; Zheng et al., 2021; Kong et al., 2022). Among the models
tested, the two high spatial resolution models (SMAP-L4C and CarbonSpace) distinguish forest stands, which is
not the case in the 50-km resolution models (LPJ-GUESS and FLUXCOM). Our results suggest an added value
of models accounting for forest stands since they are the only models to capture a clear decrease of $CO_2$ uptake
during winter in DBF plots (Fig. 3), which is consistent with the literature (Granier et al., 2002; Welp et al., 2007).
They are also the only models to capture the observed polynomial relationship between monthly 2 m temperature
and NEE over the DBF plots (Fig. 7). We have, however, to acknowledge that the temporal variability of $CO_2$
fluxes is poorly captured in the Davos ENF site, even in the SMAP-L4C and CarbonSpace models. The main
reason involves the atypical behavior of $CO_2$ fluxes in mountainous regions. Additional sites would be needed to



further understand the different responses of $CO_2$ fluxes to climate under DBF and ENF plots, which is out of the
scope of this study.

Last, there is a hiatus in the literature regarding the emergence of trends in NEE. Some studies suggest a recent
decline of $CO_2$ uptake by forest ecosystems in Europe (Smith et al., 2020; Thompson et al., 2020; Chuine et al.,
2023; van der Woude et al., 2023), while some others suggest no trend in either the recent decade or in climate
projections (Ahlström et al., 2012; Abdalla et al., 2013; Tang et al., 2013; Kong et al., 2022; Martinez del Castillo
et al., 2022; Li et al., 2023). The hiatus may be explained by the location of the sites or regions under study, (ii)
the limited temporal depth of observations (and models), (iii) whether or not these sites/regions have been affected
by wildfires and diseases (e.g., bark beetles) and (iv) whether or not wildfires and diseases are accounted for by
the models. Our results are more nuanced. We found that the evolution of the NEE annual budget does not depict
any trend but that GPP and RECO may have increased recently in the observations and some models.

**5 Conclusion**

This study questions the strengths and limitations of state-of-the-art data-driven and process-based models to
monitor and understand the temporal variability $CO_2$ exchanges between the atmosphere and western European
temperate forest ecosystems where no flux tower measurements are available. Output from two data-driven models
(CarbonSpace and FLUXCOM using different AI algorithms) and two process-based models (LPJ-GUESS and
SMAP-L4C) are inter-compared over two non-instrumented sites (Châtillonnais and Mont Beuvray, France) and
compared to $CO_2$ flux measurements from three flux tower sites (Davos, Fontainebleau and Hesse) from the
FLUXNET network retained due to their proximity with the non-instrumented sites in terms of location, climate
and forest stand. The focus is put on the representation of the annual cycle, annual budget, interannual variability
and long-term trend in $CO_2$ fluxes (NEE, GPP and RECO), as well as their relationship with various climate
parameters. Our results indicate that no model systematically outperforms the others. The best model in terms of
representing the mean annual cycle and annual budget is not necessarily the best in capturing interannual
variability. Overall, the data-driven models perform best in representing the $CO_2$ flux mean annual cycle and
annual budget, despite considerable uncertainties from one approach to another (CarbonSpace versus
FLUXCOM). As far as interannual co-variability with climate is concerned, the best performing models are those
forced by dynamic instead of static climate conditions. Our results suggest that the spatial resolution of the climate
drivers is likely very important in capturing spatial and temporal patterns in $CO_2$ exchange (e.g., in complex
mountain areas). The ability to distinguish PFT spatial heterogeneity is only partially effective in representing this.
Our results finally point towards the need to choose the appropriate model and spatial resolution according to the
scientific question to deal with and to develop high spatial resolution models forced by dynamic climate conditions
to allow for a fine scale representation of $CO_2$ flux temporal dynamics at the territorial level.



## Appendices

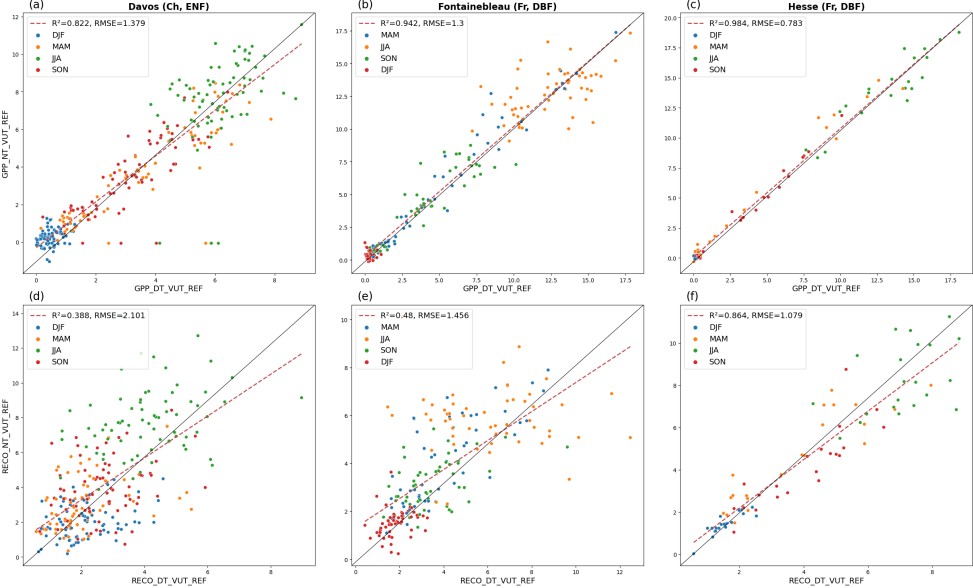

**Figure A1:** Comparison between (a-c) GPP and (d-f) RECO using the daytime partitioning (x-axis) and the nighttime partitioning (y-axis) for the three FLUXNET sites at the monthly timescale. The four colors correspond to the four seasons. The red line shows the linear regression between the two approaches, together with the coefficient of determination ($R^2$) and root mean squared error (RMSE) labeled in the insert. The black line shows the 1-by-1 correspondence.



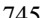

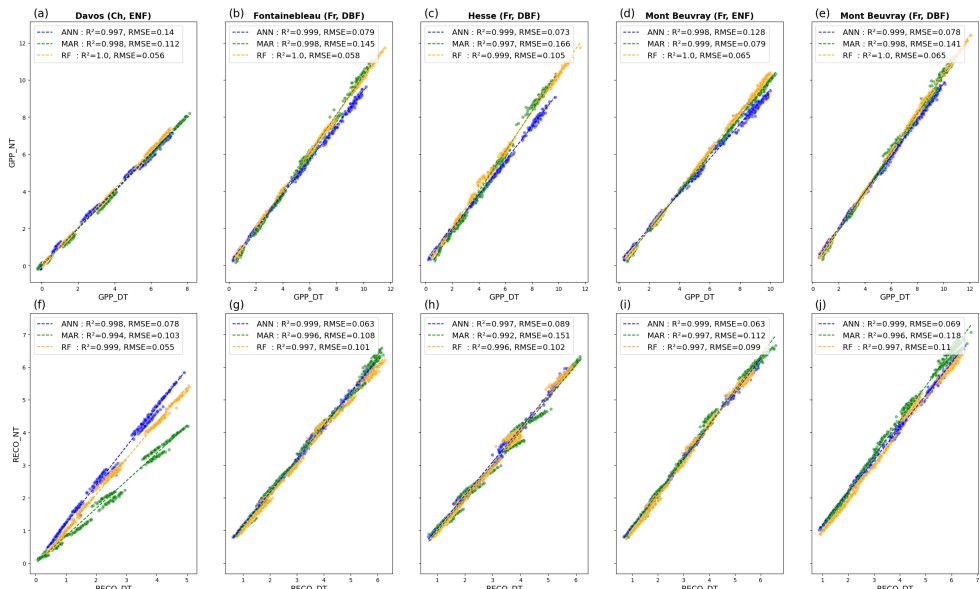

**Figure A2:** Comparison between FLUXCOM-simulated (a-e) GPP and (f-j) RECO using the daytime partitioning (x-axis) and the nighttime partitioning (y-axis) for the three FLUXNET sites at the monthly timescale. The three colors correspond to the three artificial intelligence algorithms. The colored lines show the linear regression between the two approaches, together with the $R^2$ and RMSE metrics labeled in the insert.



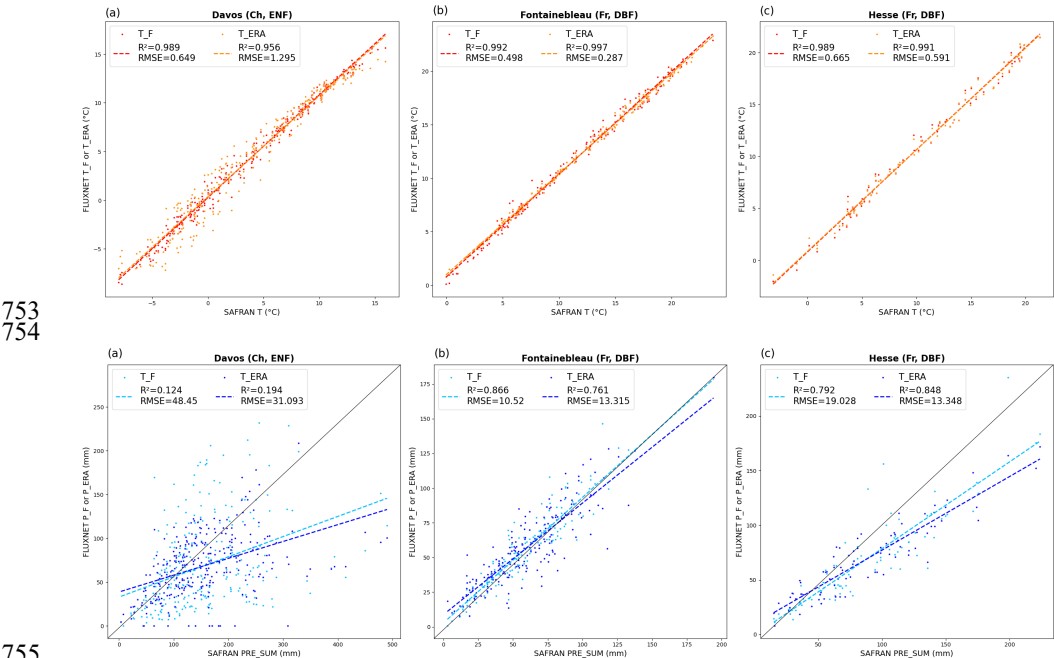



**Figure A3:** Comparison between the SAFRAN-SIM2 reanalysis (x-axis) and the FLUXNET observations (y-axis)
for (a-c) 2 m temperature and (d-f) total precipitation for the three FLUXNET sites at the monthly timescale. The
SAFRAN-SIM2 data correspond to the nearest grid point to each FLUXNET site. The SAFRAN-SIM2 –
FLUXNET comparison is done using the raw and ERA-INTERIM-corrected observations, labeled T_F/P_T and
T_ERA/P_ERA, respectively. The colored lines show the linear regression between the two datasets, together with
the coefficient of determination ($R^2$) and the root mean square error (RMSE) labeled in the insert.





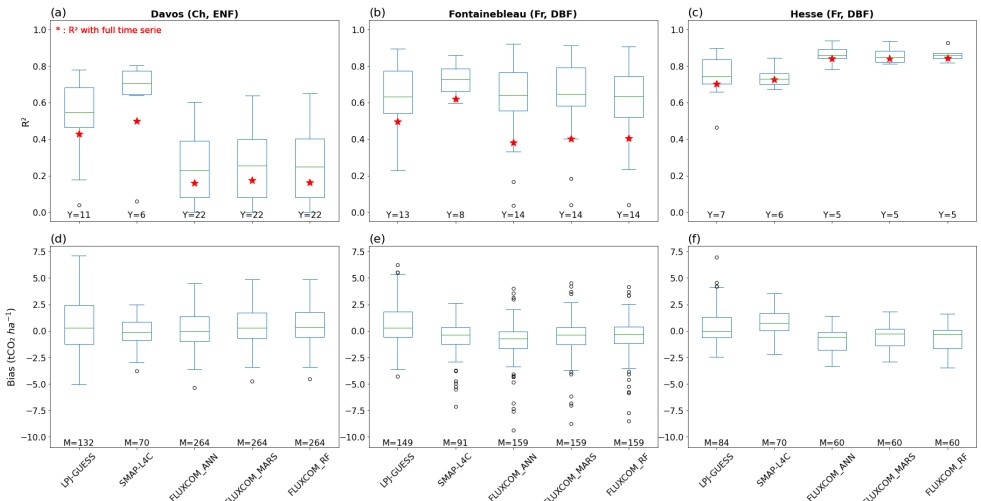

**Figure A4:** Same as Fig. 5 but for RECO.





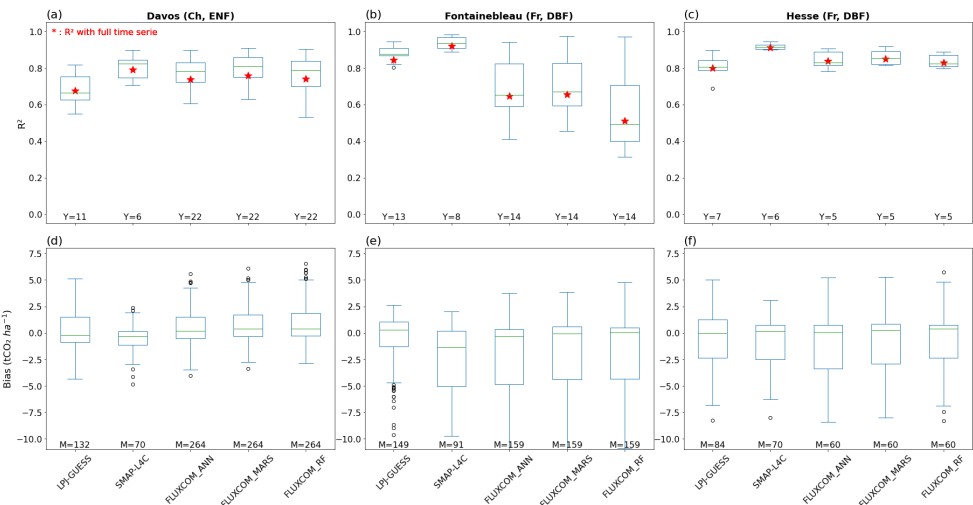

**Figure A5:** Same as Fig. 5 but for GPP.



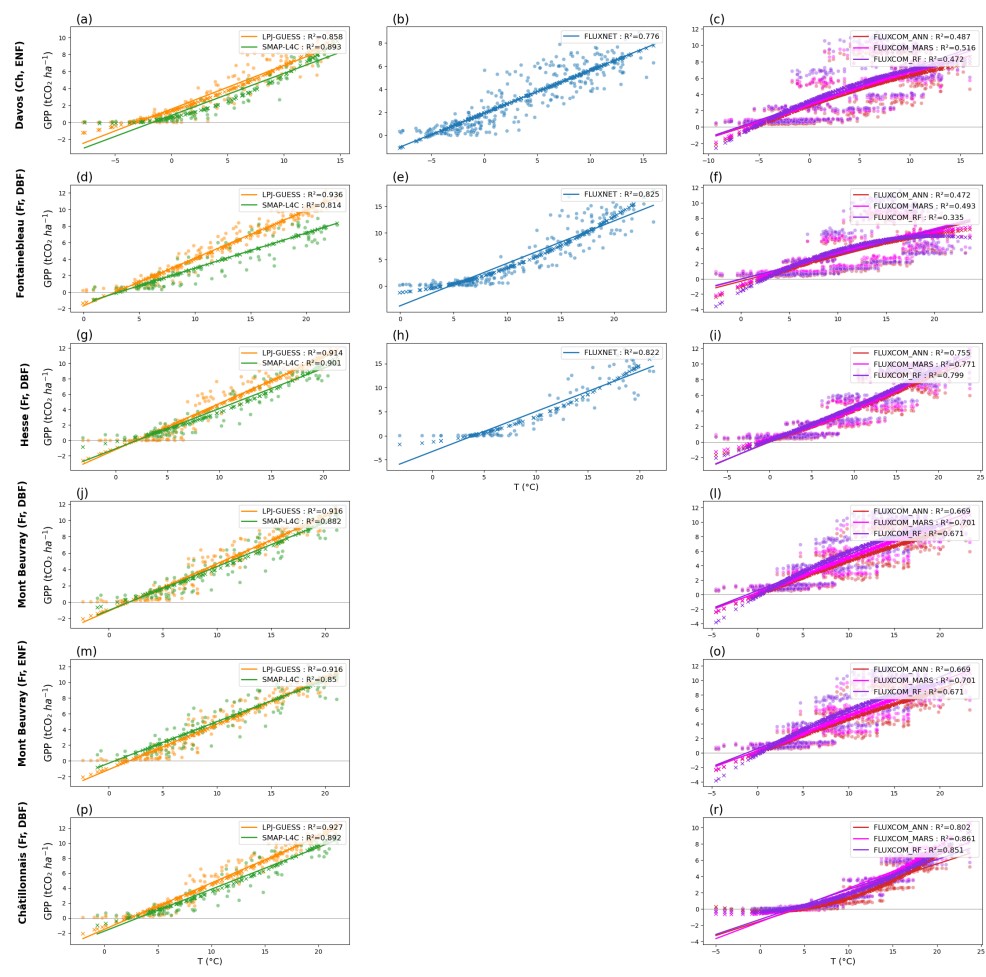

**Figure A6:** Same as Fig. 5 but for GPP.





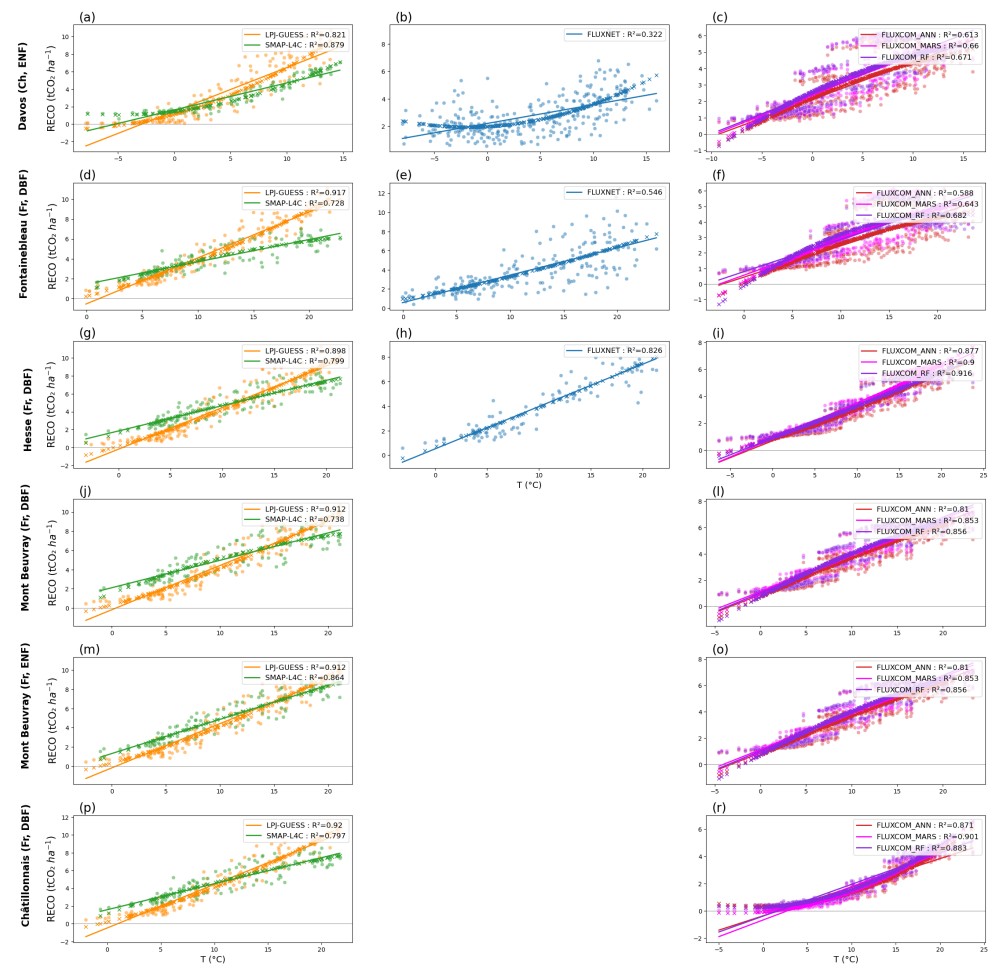

**Figure A7:** Same as Fig. 5 but for RECO.



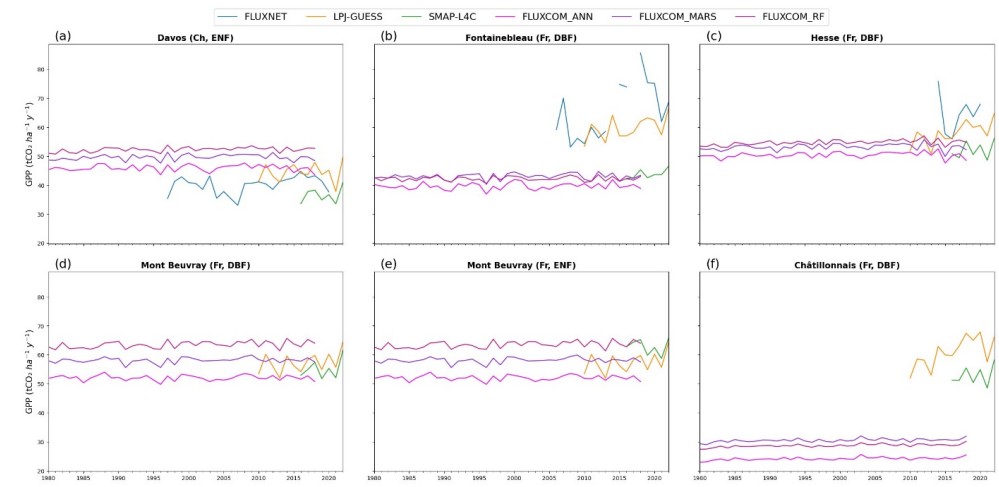



**Figure A8:** Same as Fig. 15 but for GPP.



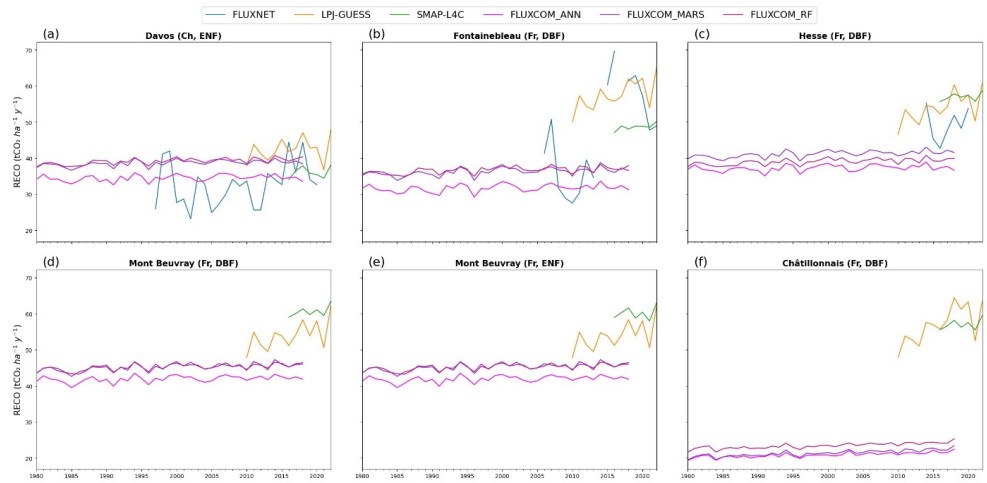



**Figure A9:** Same as Fig. 15 but for RECO.



**Acknowledgement**

This work is funded by the French National Research Agency (ANR-22-CPJ2-0026-01). We thank Daniel
Berveiller for useful discussions on eddy covariance measurements.


**Conflict of Interest**

The authors declare no competing interests.


**Data Availability**

Climate parameters from the SAFRAN-SIM2 are available at https://meteo.data.gouv.fr. $CO_2$ fluxes from the
FLUXOM data-driven model are available at https://www.bgc-jena.mpg.de. Those from LPJ-GUESS and SMAP-
L4C process-based models are avaibable at https://meta.icos-cp.eu/collections/NZNSUglRn0VeXmGDovuVY0ec
and https://nsidc.org/data/spl4cmdl/versions/7, respectively.

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
