# Peer review of "Climate impact on mean annual cycle and interannual"

_EGUsphere, 2024_

## Author Comment (AC1)

**Response to Reviewer #1**

The study evaluated $CO_2$ flux outputs from models at six sites and their relationship with climate in temperate forests of Western Europe. Overall, this manuscript addresses an important topic and presents some interesting findings. However, I am not entirely convinced that the results significantly advance our understanding of $CO_2$ fluxes at forest sites across different temporal resolutions (such as monthly and annual, as analyzed in this study).

We would like to thanks Reviewer 2 for their constructive feedback, which will greatly contributed to the improvement of the manuscript. In response to both reviewers' comments, we have decided to enhance the study by including approximately 17 additional ICOS sites, incorporating more models (specifically the TRENDY ensemble and FLUXCOM-X), and focusing exclusively on the monthly timescale. This adjustment will enable a more detailed exploration of the influence of climate on the temporal variability of $CO_2$ fluxes.

In the initial version of the manuscript, this influence was analyzed by considering all months together, which yielded results that closely resembled those at the annual timescale (e.g., see Figs. 6 and 8). In the revised version, we will examine the influence of climate on $CO_2$ fluxes for each month individually, as we anticipate that this influence will vary throughout the annual cycle. The inclusion of additional sites will also allow us to analyze the zonal and meridional variations in this influence across Europe.

General comments
1. The temporal variability in this study is not adequately addressed and may need reorganization. For example, there are ten figures illustrating monthly timescale results. However, these findings are neither included nor highlighted in the abstract or conclusion. Although there are similar patterns between the monthly and annual results, as shown in Figures 6 and 8, the monthly findings should be incorporated. Alternatively, I suggest removing or reducing the figures or text about the monthly results and focusing on annual and interannual scales.

The results are comparable at both the monthly (when all months are considered collectively) and annual timescales, which justifies the removal of all analyses conducted at the annual timescale. The results section will be reorganized into three parts: (1) mean annual cycle and interannual variability, influence of climate (2) regardless of the annual cycle (i.e., all months considered collectively) and (3) along the annual cycle (i.e. each month or season considered individually). The abstract and conclusion will be rewritten accordingly.

2. While I am not an expert in statistics, the R and $R^2$ results in this study (such as those in Figures 6 and 7) seem very close. Is it necessary to present Figure 7 in the main text? Regarding the long-term evolution, which method was used? Why do the authors state that it 'does not depict any trend'? Please provide this information in the Methods section.

R and R² provide complementary insights. R indicates the sign and strength of the relationship, while $R^2$ represents the proportion of variance in a given variable (e.g., NEE) that can be explained by another variable (e.g., 2 m temperature).

Although Figure 7 offers valuable information on temperature-induced threshold effects, it would occupy too much space with the inclusion of new sites and models. Therefore, we will remove Figure 7 in the revised manuscript.

We did not assess the statistical significance of the trends due to the limited temporal coverage of the data, particularly for observations and SMAP-L4C. As previously mentioned, we will eliminate all analyses conducted at the annual timescale, including the trend in the annual $CO_2$ flux budget.

3. The introduction of this study is not well-articulated. The discussion of the knowledge gap lacks a solid basis. Why is a finer spatial scale important? Why did the authors focus on monthly and annual timescales? The second objective, which involves the climate relationship, lacks motivation regarding the choice of variables in this study. Why were radiation or PPFD not considered?

The introduction will be revised to address the knowledge gap concerning the influence of climate on the interannual variability of $CO_2$ fluxes in European forests, as well as the need for high-resolution products. Additionally, we will provide a more robust justification for the selection of the variables included in the study.

4. It is not appropriate to assess flux trends using the FLUXCOM dataset, as FLUXCOM does not account for $CO_2$fertilization effects. Please refer to the following paper: *Jung, M., Schwalm, C., Migliavacca, M., Walther, S., Camps-Valls, G., Koirala, S., ... & Reichstein, M. (2020). Scaling carbon fluxes from eddy covariance sites to globe: synthesis and evaluation of the FLUXCOM approach. Biogeosciences, 17(5), 1343-1365.*

Thank you for this remark. We believe that $CO_2$ fertilization effects are just one of many factors influencing long-term trends in the $CO_2$ flux budget (e.g., fire, timber extraction). Consequently, this analysis will be removed in the revised version of the manuscript.

Line-by-line comments:

Line 42: "..the best models…" sounds inaccurate. Perhaps "better models" would be more appropriate, as the comparison is only made with process-based models.

Changed as suggested.

LL 45-48: could you please explain why?

At the interannual timescale, the relationship between $CO_2$ flux and climate is weaker for NEE compared to GPP and RECO. Since NEE is defined as the difference between GPP and RECO, its interannual variability arises from various combinations of these two components. For instance, positive NEE anomalies (indicating reduced $CO_2$ uptake) can occur due to a greater increase in RECO compared to GPP, a larger decrease in GPP than in RECO, a decrease in GPP with no change in RECO, or an increase in RECO without any change in GPP. This complexity suggests that the influence of climate on NEE is less direct and likely more complex than its effects on GPP and RECO.

LL 45-47: The sentence contains a lot of information and lacks clarity. I suggest revising it for better readability.

We will revise this sentence for better readability.

Line 49: How long is the 'long-term'?

The term "long-term" is contingent on the length of data availability, which does not exceed 20 years. This is why we have placed the term in quotation marks. However, this remark will no longer apply in the revised manuscript, as analyses conducted at the annual timescale will be removed.

LL 50-53: The statement is too general and does not seem related to the main point of the study.

Thank you for this remark. This statement will be removed in the revised manuscript.

Line 61: 'At the global scale, forest ecosystems cover about 30% of landmasses' suggest adding a reference.

We will add this reference:
FAO Global Forest Resources Assessment 2020: Terms and Definitions. For. Resour. Assess. Work. Pap.2020, 32

LL 71-72: What is the meaning of this number here?

According to Chuine et al. (2023) timber-extraction and climate-related mortality have increased by 20% and 54% between 2005-2013 and 2012-2020, respectively.

Line 101: Suggest deleting 'when available'

Done as suggested.

LL 101-113: Suggest moving these sentences to another paragraph or placing them above the objectives.

Thank you for the suggestion. We will move these sentences above the objectives.

LL127-128: Redundace, and which varaible? all fluxes?

This sentence was unclear. It will be removed in the revised version.

Line 132: "Extreme" can refer to both high and low conditions. To specify, consider using "extreme high" or "extreme hot" for clarity.

Thank you for this remark. Done as suggested.

LL 132-133: Suggest revising the sentence.

The sentence has been revised. It now reads: "In addition, extreme high temperatures combined with soil water stress have been shown to significantly impact GPP and RECO".

LL147-149: Suggest deleting these sentences.

Done.

Figure 1: If the reader is not familiar with Europe, it may not be clear. Consider adding specific locations, such as "France".

Figure 1 will be modified to include the new sites.

Table 1: The table format needs adjustment. Please refer to the journal's guidelines, which typically suggest using horizontal lines only above and below the table, and as a separator between the table header and the main body.

Table 1 will be adjusted to follow the journal's guidelines.

Figure 9-11: Perhaps move to supplement.

Figures 9 to 11 will be moved to supplement.

LL555-556: Suggest deleting the sentence.

We believe it is important to retain this information in the figure caption to help readers understand why the results are similar between ENF and DBF.

Line 603: "qualitatively similar" might sound speculative?

You are correct. We will mention that the results obtained at the annual timescale are similar to those at the monthly timescale. A few results from the annual timescale will be included in the supplementary material to support this statement.

Line 724: Replace 'best' to 'better'.

Done.

Line 772: Should be 'Fig. 7' not 'Fig, 5'

You are correct.

Line 776: 'Fig. 7'

Done.

---

## Author Comment (AC2)

**Response to Reviewer #2**

This study evaluates the performance of various carbon flux products against eddy covariance measurements at three forest sites in France. The authors investigate the monthly, seasonal, and inter-annual variability of NEE, GPP, and RECO to assess different global products and explore their relationships with meteorological variables. While the manuscript is generally written clearly, the analysis lacks sufficient depth and significance for the scientific community. This isn't to say that evaluating existing products isn't valuable, but the limited number of eddy covariance sites and the selection of only four global products raise concerns about the comprehensiveness of the study.

We would like to thanks Reviewer 2 for their constructive feedback, which will greatly contributed to the improvement of the manuscript. In response to both reviewers' comments, we have decided to enhance the study by including approximately 17 additional ICOS sites, incorporating more models (specifically the TRENDY ensemble and FLUXCOM-X), and focusing exclusively on the monthly timescale. This adjustment will enable a more detailed exploration of the influence of climate on the temporal variability of $CO_2$ fluxes.

In the initial version of the manuscript, this influence was analyzed by considering all months together, which yielded results that closely resembled those at the annual timescale (e.g., see Figs. 6 and 8). In the revised version, we will examine the influence of climate on $CO_2$ fluxes for each month individually, as we anticipate that this influence will vary throughout the annual cycle. The inclusion of additional sites will also allow us to analyze the zonal and meridional variations in this influence across Europe.

1.Choice of Models (LPJ-GUESS and FLUXCOM v1): Why were LPJ-GUESS and FLUXCOM v1 selected when many other land surface models or upscaled products are available for evaluation? Including more models and products could improve relevance, especially since ensemble models are commonly used in land carbon sink studies.

The paper does not aim to evaluate all existing $CO_2$ flux models. One objective is to assess the strengths and limitations of the CarbonSpace data-driven model in comparison to several widely used models. The CarbonSpace model is distinctive due to its very high spatial resolution and its ability to differentiate between tree species.

We acknowledge that the use of a single land surface model (LPJ-GUESS) was not ideal, given the large spread across land surface models. To address this, we will incorporate the TRENDY v12 ensemble (S3 simulations, which include time-varying $CO_2$, climate, and land use) to better account for uncertainties in process-based models.

The choice of FLUXCOM remains appropriate, as it is frequently referenced in the literature. However, we will consider adding the FLUXCOM-X product, which provides global monthly $CO_2$ data at an unprecedented spatial resolution of 0.05°.

2.Spatial Resolution Mismatch: The 50 km spatial resolution of LPJ-GUESS and FLUXCOM products may not align with the footprint of the eddy covariance sites, and this mismatch is not addressed in the manuscript.

Comparing coarse-resolution models with local measurements can indeed result in significant discrepancies that may not necessarily reflect model errors. While we do not anticipate a one-to-one match, we expect that the main observed patterns to be captured by both data-driven and process-based models. It is also important to note that all the data-driven models used in this study are trained using FLUXNET2015 local measurements, making comparisons between CarbonSpace/FLUXCOM estimates and local measurements meaningful. We will include a brief discussion on the impact of spatial resolution mismatches.

3.Insufficient Number of Sites: Only three forest sites are used in the analysis, despite the availability of hundreds of FLUXNET sites globally. Using only these sites may not provide a robust basis for summarizing product performance.

The initial aim of this paper was to evaluate data-driven and process-based models to capture the annual cycle, interannual variability, and trends of $CO_2$ fluxes in temperate deciduous broadleaf (DBF) and evergreen needleleaf (ENF) forests of Western Europe. We recognize that the number of sites analyzed was insufficient. To address this, we will expand the analysis to encompass all of Europe. This expansion will allow us to include 17 additional ICOS sites that were not part of the initial version, resulting in a total of 20 ICOS sites under study. Each of the 20 ICOS sites provides at least five years of data (Figure 1).

[Figure]

Figure 1: ICOS sites providing $CO_2$ fluxes for at least 5 years.

4.Correlation Analysis: The correlation analysis in Figure 6 lacks a logical basis, as some variables (e.g., VPD and RECO) do not have clear biogeochemical or biophysical relationships. Also, the analysis does not account for multicollinearity among variables, which affects the validity of the results.

While we recognize that the physical influence of VPD on GPP and NEE is more evident than on RECO, it is important to note that autotrophic respiration is closely correlated with GPP. As a result, VPD indirectly influences RECO through its impact on GPP.

The other meteorological variables (2 m temperature, soil moisture, total precipitation, and real and potential evapotranspiration) exert a significant, physically grounded influence on NEE, GPP and RECO, justifying their inclusion in the analysis.

While multicollinearity poses a strong issue when analysing the combined influence of climate variables on $CO_2$ fluxes (e.g., in multiple linear regression), this study computes correlations variable by variable, which is methodologically sound and unaffected by multicollinearity.

5.Focus on Temperature: The authors only consider temperature when analyzing carbon fluxes in Figure 6 and do not include other important variables, such as soil moisture, which were emphasized in the introduction. Given this, the use of polynomial regressions without considering other factors raises questions about interactive effects of multivariate factors of the carbon fluxes.

The polynomial regressions shown in Figure 7 are valuable for discussing temperature-induced threshold effects, known to affect both GPP and RECO. However, with the inclusion of new models and additional sites, Figure 7 would occupy too much space. Therefore, it will be omitted from the revised manuscript.

Specific Points:

Inconsistent Visualization (Figures 5 and 13): Figures 5 and 13 present similar data for annual and monthly scales, but the visualizations need to be consistent to enable direct comparison.

Figure 13 will be removed in the revised manuscript to maintain a focus solely on the monthly timescale and further investigate the climate – $CO_2$ flux relationship along the annual cycle.

Climate Anomalies Definition: The authors should clarify how climate anomalies are defined, as the methods section only explains CO2 flux anomalies. Also, the choice of the -0.5/+0.5 thresholds for carbon flux anomalies seems arbitrary and needs further justification.

Climate anomalies are computed in the same manner as $CO_2$ flux anomalies, resulting in variations in the reference period depending on the site. With regard to the composite analysis, higher thresholds (-1/+1 and -1.5/+1.5) significantly limit the sample size due to the short duration of the data. Therefore, we have opted for -0.5/+0.5 thresholds as a compromise. In the revised manuscript, we will provide a more detailed explanation of how

climate anomalies are computed and justify our choice of thresholds in Section 2.4 (Methodology).

---

## Author Response (AR1)

**Response to Associate Editor**

We've received two reviews of the manuscript. One suggests major revisions and the other recommends rejection. Upon reviewing the manuscript myself, I find the objectives to be interesting and the analysis to be expansive for validating and intercomparing several model types. Nevertheless, the reviewers raised some constructive points about the novelty and presentation of results that I agreed with nearly all. Therefore, we request substantial revisions addressing all comments. I've added a few overarching points to address giving context to the referee points.

We sincerely appreciate the opportunity to improve our paper. Thanks to the insightful comments from the reviewers and yourself, we have significantly strengthened the study by incorporating all available ENF and DBF sites in Europe with at least five years of data, integrating additional data-driven and process-based models, and refining our objectives. The paper now focuses on the space-time variability of $CO_2$ fluxes across European forests and investigates how climate influences both their annual cycle and interannual variability.

1) The site selection needs more motivation than lines 154-156. The introduction motivates the number of FLUXNET and ICOS sites around line 80, but then only uses a few specific sites.

We no longer provide the total number of sites available in the FLUXNET database, as this global-scale information was misleading. Instead, we now report the number of flux towers located in Europe, which is 98 across all ecosystem types (see l122-124). This number decreases significantly when focusing on the ENF and DBF land cover classes, with only 24 sites. Among these, we selected 19 ENF sites, all of which have at least five years of data.

The site selection and characteristics are now fully justified in section 2.1. It now reads (l163-173): "Out of the 24 ENF and DBF sites from the ICOS network, we selected the 19 sites (Fig. 1), 13 classified as ENF and 6 as DBF, for which observed $CO_2$ fluxes are available for at least 5 years (Table 1). These sites allow to sample the different climatic zones of Europe. Three ENF sites (FR-Bil, FR-FBn and IT-SR2) are located in the northern region of southern Europe, close to 45°N, and ranging from sea level to 400 m in elevation. They are characterized by mild, wet winters and hot dry summers, with annual mean temperature and precipitation of 12.9–13.90 °C and 700–960 mm, respectively. Four ENF sites (FI-Hyy, FI-Let, SE-Nor and SE-Svb) are located in northern Europe (60-65°N) at an elevation below 270 m. They are characterized by subarctic climate with annual mean temperature and precipitation of 1.8–6.5 °C and 586–711 mm, respectively. The remaining twelve sites (6 DBFs and 6 ENFs) are situated in central Europe within the 45–60°N, 2.5–20°E domain, encompassing a wide range of elevations (40–1730 m) and spanning temperate to continental climates. As a result, they exhibit substantial variability in annual mean temperature (4.3–11.4 °C) and precipitation (563–1338 mm)".

2) Both referees found the presentation of results to be fair or poor and I agree there are likely better methods to plot the data. For example, it would be nice to call out more clearly the FLUXNET data in the results figures. The color is similar to the others and not easy to quickly

find. Please prioritize means to improve the readability of the figures such as increasing font size, or reducing panels if possible.

The results are now presented much more clearly in the revised manuscript. We first describe the mean annual cycle and interannual variability of observed $CO_2$ fluxes (Section 3.1), followed by an evaluation of the models' ability to capture these observed patterns (Section 3.2). Finally, we examine the impact of climate on both the annual cycle and month-by-month interannual variability of $CO_2$ fluxes (Section 3.3).

The figures have also been improved. While the inclusion of additional sites and datasets means that some figures are not necessarily simpler than before and require a full-page width for readability (Figs. 5 and 9), they provide a more comprehensive synthesis of the space-time variation in $CO_2$ fluxes and the influence of climate on these dynamics.

3) I agree with referee 1 that the introduction could go a bit further to be more specific about the knowledge gaps being addressed.

The introduction has been completely rewritten to better reflect current understanding of how climate influences the temporal variability of $CO_2$ fluxes. We have also expanded our discussion of the knowledge gaps addressed by our study. It now reads (l142-148): "Most recent studies examining the influence of climate on the temporal dynamics of European forest $CO_2$ fluxes rely on case studies and primarily focus on spring and summer conditions (Smith et al., 2020; Thompson et al., 2020; van der Woude et al., 2023). However, a more comprehensive assessment is needed across the entire annual cycle, as $CO_2$ release during fall and winter is expected to increase under climate change. Additionally, climate conditions vary significantly between northern and southern Europe, necessitating a broader spatial perspective. These objectives are addressed at the monthly timescale, which is considered sufficiently fine to capture both the $CO_2$ flux annual cycle and its interannual variability".

4) There are some limitation points that do need mention here. First, the spatial scale difference between the flux tower and model resolution as mentioned by a referee. Second, we need to acknowledge the data driven approaches may be trained partly on these flux towers and there may then be concern about out-of-sample testing (or performance on a new site it wasn't trained on).

The first limitation (spatial mismatch) is now fully discussed. It now reads (l625-631): "As a first step, we assess the model abilities to reproduce the observed mean annual cycle and interannual variability of $CO_2$ fluxes. This evaluation presents two key challenges. First, the temporal coverage of observations in the FLUXNET database is often limited, making it difficult to extract robust signals, particularly for interannual variability. Second, there is a spatial scale mismatch between site-level observations, representing fluxes from the tower footprint to several square kilometers (Göckede et al., 2008), and most models used in this study, which simulate fluxes at regional to large scales, except for the hectometric-scale CarbonSpace model. Given these constraints, our evaluation should be considered qualitative rather than strictly quantitative".

The objective of the revised manuscript is no longer solely to evaluate the models but rather to use them as complementary tools for gaining deeper insights into the impact of climate on the temporal dynamics of $CO_2$ fluxes, particularly in terms of interannual variability. The superior performance of data-driven models compared to process-based models is no longer a concern, as our focus is not on determining which model performs better. Instead, data-driven models provide a unique opportunity to investigate climate impacts on $CO_2$ flux dynamics, as they incorporate most, if not all, ICOS sites.

We have added a few sentences to acknowledge that data-driven models are expected to outperform process-based models. It now reads:
- l249-255: "The four data-driven models include most, if not all, ICOS sites mobilised in this study. They accurately capture the mean annual and seasonal cycles of $CO_2$ fluxes (Tramontana et al., 2016; Jung et al., 2020; He et al., 2022; Zhuravlev et al., 2022) and are expected to outperform process-based models since the latter do not assimilate observed $CO_2$ fluxes. The methodological framework (e.g., machine learning model, forcing data and horizontal resolution) remains different between the data-driven models. An inter-model convergence will be interpreted as a forced response imposed by the observations. A divergence will be interpreted as uncertainties induced by the methodological framework";
- l640-647: "The interannual variability in the models is weaker than in the observations, consistent with previous studies (e.g., Nelson et al., 2024). Yet, the co-variability between observed and simulated $CO_2$ fluxes remains correct despite the aforementioned constraints. This agreement was expected for data-driven models, as they incorporate FLUXNET observations in their development. However, it was less anticipated for process-based models, which do not assimilate direct $CO_2$ flux measurements. Their ability to capture observed interannual variability likely stems from the fact that TRENDY models are driven by observed $CO_2$ concentrations, land-use changes, and climate data, while the SMAP-L4C model benefits from the assimilation of satellite-derived soil moisture observations".

5) I think the use of monthly data to look at the "annual cycle" is acceptable. However, since the reviewer questioned this point, it might be good to reiterate in the results/discussion that the monthly results are focused on the annual cycle.

We have clarified throughout the revised manuscript (and the figure captions) that the study aims to assess the impact of climate on the annual cycle and interannual variability of monthly $CO_2$ fluxes in European DBFs and ENFs:
- l147-148: "These objectives are addressed at the monthly timescale, which is considered sufficiently fine to capture both the $CO_2$ flux annual cycle and its interannual variability";
- l316-317: "Since these datasets have varying temporal resolutions (Tables 1 and 2), all were aggregated to a monthly timescale";
- l378-379: "Figure 4a displays the mean annual cycle of monthly NEE, GPP and RECO as provided by FLUXNET observations";
- l485: "Figure 7 shows the correlations between the modelled and observed interannual variability in monthly NEE";

- l621-623: "This study aims at assessing the impact of climate on annual cycle and interannual variability of monthly $CO_2$ fluxes in European DBFs and ENFs through conjointly analysing observations from the FLUXNET network and state-of-the-art data-driven and process-based models".
- l695-697: "This study makes use of state-of-the-art data-driven and process-based models to complement observations for assessing the impact of climate on the annual cycle and interannual variability of monthly $CO_2$ fluxes in European DBFs and ENFs".

**Response to Reviewer #1**

The study evaluated $CO_2$ flux outputs from models at six sites and their relationship with climate in temperate forests of Western Europe. Overall, this manuscript addresses an important topic and presents some interesting findings. However, I am not entirely convinced that the results significantly advance our understanding of $CO_2$ fluxes at forest sites across different temporal resolutions (such as monthly and annual, as analyzed in this study).

We would like to thanks Reviewer 1 for their constructive feedback, which will greatly contributed to the improvement of the manuscript. In response to both reviewers' comments, we have enhanced the study by focusing on 19 ICOS sites (instead of the 4 initial ICOS sites), incorporating more models (specifically the TRENDY, FLUXCOM-ERA5 and FLUXCOM-MODIS ensembles together with the newly released FLUXCOM-X), and focusing exclusively on the monthly timescale. This adjustment enables a more detailed exploration of the influence of climate on the temporal variability of $CO_2$ fluxes.

In the initial version of the manuscript, this influence was analyzed by considering all months together, which yielded results that closely resembled those at the annual timescale (e.g., see old Figs. 6 and 8). In the revised version, we examine the influence of climate on $CO_2$ fluxes for each month individually and show that this influence varies throughout the annual cycle and across Europe.

Based on Reviewer 1's comments, we have revised the study's objective to focus on assessing how climate influences the annual cycle and interannual variability of $CO_2$ fluxes, while examining how this relationship varies throughout the year and across Europe. To achieve this, we jointly analyse observations, whose temporal coverage remains limited, alongside state-of-the-art data-driven and process-based models, after ensuring that these models adequately capture the observed patterns. Additionally, we have updated the title to better reflect the revised objective. The paper is now entitled "Climate impact on mean annual cycle and interannual variability of CO2 fluxes in European DBF and ENF forests: insights from observations and state-of-the-art data-driven and process-based models".

General comments
1. The temporal variability in this study is not adequately addressed and may need reorganization. For example, there are ten figures illustrating monthly timescale results. However, these findings are neither included nor highlighted in the abstract or conclusion. Although there are similar patterns between the monthly and annual results, as shown in Figures 6 and 8, the monthly findings should be incorporated. Alternatively, I suggest removing or reducing the figures or text about the monthly results and focusing on annual and interannual scales.

The revised manuscript focuses exclusively on the monthly timescale. First, we evaluate the models' ability to reproduce the mean annual cycle and interannual variability of observed $CO_2$ fluxes, considering the overlapping periods between each model–observation pair. Next, we analyse the impact of climate on both the annual cycle and interannual variability of $CO_2$ fluxes. For interannual variability, we extend the analysis beyond the first version of the

manuscript by assessing climate influences for each month of the year. Additionally, we have revised the abstract to ensure all key findings are well highlighted and have improved the balance between sections.

2.  While I am not an expert in statistics, the R and $R^2$ results in this study (such as those in Figures 6 and 7) seem very close. Is it necessary to present Figure 7 in the main text? Regarding the long-term evolution, which method was used? Why do the authors state that it 'does not depict any trend'? Please provide this information in the Methods section.

R and $R^2$ provide complementary insights. R indicates the sign and strength of the relationship, while $R^2$ represents the proportion of variance in a given variable (e.g., NEE) that can be explained by another variable (e.g., 2 m temperature).

Although Figure 7 offers valuable information on temperature-induced threshold effects, it would have occupying too much space with the inclusion of new sites and models. Therefore, we removed Figure 7 in the revised manuscript.

Trends are no more examined based on Reviewer 1's comments.

3.  The introduction of this study is not well-articulated. The discussion of the knowledge gap lacks a solid basis. Why is a finer spatial scale important? Why did the authors focus on monthly and annual timescales? The second objective, which involves the climate relationship, lacks motivation regarding the choice of variables in this study. Why were radiation or PPFD not considered?

The introduction has been entirely rewritten to address the knowledge gap concerning the influence of climate on the interannual variability of $CO_2$ fluxes in European forests, as well as the need to use models to complement the observational data.

In particular, the introduction now lists the main impacting climate variables. Note that preliminary analyses show that the impact of incident shortwave radiation on $CO_2$ flux annual cycle and interannual variability is similar to that of 2 m temperature. We thus consider only 2 m temperature in this study, which affects both GPP and RECO and NEE. This is now stated in section 2.2.4.

Finally, note that we did not use PPFD, as this variable is unavailable in the ERA5-Land product, which replaces the France-centered SAFRAN-SIM2 reanalysis in the revised manuscript.

4.  It is not appropriate to assess flux trends using the FLUXCOM dataset, as FLUXCOM does not account for $CO_2$ fertilization effects. Please refer to the following paper: *Jung, M., Schwalm, C., Migliavacca, M., Walther, S., Camps-Valls, G., Koirala, S., ... & Reichstein, M. (2020). Scaling carbon fluxes from eddy covariance sites to globe: synthesis and evaluation of the FLUXCOM approach. Biogeosciences, 17(5), 1343-1365.*

Thank you for this remark. We believe that $CO_2$ fertilization effects are just one of many factors influencing long-term trends in the $CO_2$ flux budget (e.g., fire, timber extraction). However, we no longer analyse long-term trends in this study.

Line-by-line comments:

Please note that most of the line-by-line comments are no longer applicable, as the revised manuscript has been almost entirely rewritten.

Line 42: "..the best models…" sounds inaccurate. Perhaps "better models" would be more appropriate, as the comparison is only made with process-based models.

No more applies.

LL 45-48: could you please explain why?

This points is now addressed in the discussion section. It now reads (l665-671): "Regarding interannual variability, the climate impact on CO2 fluxes can be summarized in three points. First, climate impacts more strongly GPP and RECO than NEE, regardless of the site and dataset. Since NEE is the difference between GPP and RECO, its interannual variability arises from various combinations of these two components. For instance, reduced CO2 sequestration can results from a greater increase in RECO compared to GPP, a larger decrease in GPP than in RECO, a decrease in GPP with no change in RECO, or an increase in RECO without any change in GPP. Such as complexity implies that the influence of climate on NEE is less direct and likely more intricate than its effects on GPP and RECO".

LL 45-47: The sentence contains a lot of information and lacks clarity. I suggest revising it for better readability.

The abstract has been completely rewritten and is hopefully clearer in the revised manuscript.

Line 49: How long is the 'long-term'?

The term "long-term" is contingent on the length of data availability, which does not exceed 20 years. This is why we have placed the term in quotation marks. However, this remark does no longer apply in the revised manuscript, as analyses conducted at the annual timescale have been removed.

LL 50-53: The statement is too general and does not seem related to the main point of the study.

Thank you for this remark. This statement has been removed in the revised manuscript.

Line 61: 'At the global scale, forest ecosystems cover about 30% of landmasses' suggest adding a reference.

This sentence has been simplified. It now reads (l85-87): "Forest ecosystems are the largest part of the land CO2 sink (Lindeskog et al., 2021), with up to 20-50% of anthropogenic CO2 emissions (land-use changes excluded) sequestered for the 2000-2010 period (Le Quéré et al., 2018; Pugh et al., 2019; Pan et al., 2024)".

LL 71-72: What is the meaning of this number here?

According to Chuine et al. (2023) timber-extraction and climate-related mortality have increased by 20% and 54% between 2005-2013 and 2012-2020, respectively. This sentence has been removed since we no longer focus on $CO_2$ flux trends.

Line 101: Suggest deleting 'when available'

Done as suggested.

LL 101-113: Suggest moving these sentences to another paragraph or placing them above the objectives.

Thank you for the suggestion. These sentences have been placed above the objectives.

LL127-128: Redundace, and which varaible? all fluxes?

This sentence has been removed. We now clearly identify which variable impacts on each $CO_2$ flux. It now reads (l95-105): "Numerous studies have demonstrated the strong influence of climate on $CO_2$ exchanges between the atmosphere and forest ecosystems. The annual cycle, and to a lesser extent, interannual variability of these fluxes, are driven by factors such as incident shortwave radiation, temperature, atmospheric evaporative demand, and the water cycle, including soil moisture dynamics (Haszpra et al., 2005; Tang et al., 2014; von Buttlar et al., 2018; Kong et al., 2022; Sharma et al., 2022; Li et al., 2023; Xu et al., 2023). The dominant climate factor influencing $CO_2$ fluxes depends on the specific component considered. The variability in net ecosystem exchanges (NEE) is a mixed response of its two components: gross primary production (GPP), which sequesters $CO_2$ into the ecosystem through photosynthesis, and ecosystem respiration (RECO), which releases $CO_2$ into the atmosphere from forest metabolism (autotroph respiration) and the decomposition of organic matter by fungi and bacteria (heterotrophic respiration). GPP is primarily driven by vapour pressure deficit (VPD), shortwave radiation, temperature, and soil moisture, while RECO is mainly influenced by precipitation, soil moisture, and temperature (Messori et al., 2019)".

Line 132: "Extreme" can refer to both high and low conditions. To specify, consider using "extreme high" or "extreme hot" for clarity.

Thank you for your remark. We no longer refer to extreme events, as the revised manuscript no longer includes results from the composite analysis.

LL 132-133: Suggest revising the sentence.

The sentence has been removed.

LL147-149: Suggest deleting these sentences.

Done.

Figure 1: If the reader is not familiar with Europe, it may not be clear. Consider adding specific locations, such as "France".

Figure 1 now displays the locations of all 19 ICOS sites included in this study, along with their IGBP classification and temporal coverage.

Table 1: The table format needs adjustment. Please refer to the journal's guidelines, which typically suggest using horizontal lines only above and below the table, and as a separator between the table header and the main body.

All Tables (1, 2 and A1) have been adjusted to follow the journal's guidelines.

Figure 9-11: Perhaps move to supplement.

We no longer show results from composite analyses. Figures 9 to 11 have been removed.

LL555-556: Suggest deleting the sentence.

Figure 12 has been removed since we no longer show annual budgets.

Line 603: "qualitatively similar" might sound speculative?

This has been removed. However, we still use the term "qualitative" in the context of model evaluation, as the limited temporal coverage of observational data prevents a robust quantitative assessment of model skill.

Line 724: Replace 'best' to 'better'.

No more applies.

Line 772: Should be 'Fig. 7' not 'Fig, 5'

No more applies.

Line 776: 'Fig. 7'

No more applies.

**Response to Reviewer #2**

This study evaluates the performance of various carbon flux products against eddy covariance measurements at three forest sites in France. The authors investigate the monthly, seasonal, and inter-annual variability of NEE, GPP, and RECO to assess different global products and explore their relationships with meteorological variables. While the manuscript is generally written clearly, the analysis lacks sufficient depth and significance for the scientific community. This isn't to say that evaluating existing products isn't valuable, but the limited number of eddy covariance sites and the selection of only four global products raise concerns about the comprehensiveness of the study.

We would like to thanks Reviewer 2 for their constructive feedback, which will greatly contributed to the improvement of the manuscript. In response to both reviewers' comments, we have enhanced the study by focusing on 19 ICOS sites (instead of the 4 initial ICOS sites), incorporating more models (specifically the TRENDY, FLUXCOM-ERA5 and FLUXCOM-MODIS ensembles together with the newly released FLUXCOM-X), and focusing exclusively on the monthly timescale. This adjustment enables a more detailed exploration of the influence of climate on the temporal variability of $CO_2$ fluxes.

In the initial version of the manuscript, this influence was analyzed by considering all months together, which yielded results that closely resembled those at the annual timescale (e.g., see old Figs. 6 and 8). In the revised version, we examine the influence of climate on $CO_2$ fluxes for each month individually and show that this influence varies throughout the annual cycle and across Europe.

Based on Reviewer 1's comments, we have revised the study's objective to focus on assessing how climate influences the annual cycle and interannual variability of $CO_2$ fluxes, while examining how this relationship varies throughout the year and across Europe. To achieve this, we jointly analyse observations, whose temporal coverage remains limited, alongside state-of-the-art data-driven and process-based models, after ensuring that these models adequately capture the observed patterns. Additionally, we have updated the title to better reflect the revised objective. The paper is now entitled "Climate impact on mean annual cycle and interannual variability of CO2 fluxes in European DBF and ENF forests: insights from observations and state-of-the-art data-driven and process-based models".

1.Choice of Models (LPJ-GUESS and FLUXCOM v1): Why were LPJ-GUESS and FLUXCOM v1 selected when many other land surface models or upscaled products are available for evaluation? Including more models and products could improve relevance, especially since ensemble models are commonly used in land carbon sink studies.

Thank you for your remark. The revised manuscript now incorporates 15 DVGMs from the TRENDY project (S3 simulations, which include time-varying $CO_2$, climate, and land use) instead of relying solely on LPJ-GUESS. This broader selection provides a more comprehensive representation of the uncertainties associated with process-based models.

Additionally, we have included:

- the FLUXCOM-MODIS ensemble, which consists of data-driven models using different machine learning methods and relying only on satellite-derived vegetation data, unlike FLUXCOM-ERA5, which includes both climate and vegetation variables as predictors ;

- the FLUXCOM-X model, a newly released and high spatial resolution (0.05°) product that improves coverage, training quality, and satellite data processing and incorporates additional sites compared to previous FLUXCOM versions.

Please note that the goal of this study is not to include all existing $CO_2$ flux models. Instead, by jointly analysing observations alongside a large ensemble of DVGMs, a reanalysis process-based model (SMAP), various FLUXCOM models, and a site-scale data-driven model (CarbonSpace), we provide a comprehensive assessment of how climate influences the annual cycle and interannual variability of European forest $CO_2$ fluxes, while also addressing the associated uncertainties.

2.Spatial Resolution Mismatch: The 50 km spatial resolution of LPJ-GUESS and FLUXCOM products may not align with the footprint of the eddy covariance sites, and this mismatch is not addressed in the manuscript.

Comparing coarse-resolution models with local measurements can indeed result in significant discrepancies that may not necessarily reflect model errors. While we do not anticipate a one-to-one match, we expect that the main observed patterns to be captured by both data-driven and process-based models. It is also important to note that all the data-driven models used in this study are trained using FLUXNET2015 local measurements, making comparisons between CarbonSpace/FLUXCOM estimates and local measurements meaningful. We now discuss the spatial resolution mismatch in section 4. It now reads (l625-631): "As a first step, we assess the model abilities to reproduce the observed mean annual cycle and interannual variability of $CO_2$ fluxes. This evaluation presents two key challenges. First, the temporal coverage of observations in the FLUXNET database is often limited, making it difficult to extract robust signals, particularly for interannual variability. Second, there is a spatial scale mismatch between site-level observations, representing fluxes from the tower footprint to several square kilometers (Göckede et al., 2008), and most models used in this study, which simulate fluxes at regional to large scales, except for the hectometric-scale CarbonSpace model. Given these constraints, our evaluation should be considered qualitative rather than strictly quantitative".

3.Insufficient Number of Sites: Only three forest sites are used in the analysis, despite the availability of hundreds of FLUXNET sites globally. Using only these sites may not provide a robust basis for summarizing product performance.

Thank you for this remark. We have significantly increased the number of sites considered in this study by expanding the study domain to Europe. Across Europe, the FLUXNET database includes 24 ENF and DBF sites, from which we selected all sites with at least five years of $CO_2$ flux data. The final selection includes 19 sites, shown in Fig. 1 and detailed in Table 1 of the

revised manuscript. This expanded dataset enables us to assess the extent to which $CO_2$ flux temporal dynamics and the influence of climate change across Europe.

4.Correlation Analysis: The correlation analysis in Figure 6 lacks a logical basis, as some variables (e.g., VPD and RECO) do not have clear biogeochemical or biophysical relationships. Also, the analysis does not account for multicollinearity among variables, which affects the validity of the results.

We recognize that the physical influence of VPD on GPP and NEE is more evident than on RECO. However, it is important to note that respiration is closely correlated with GPP. As a result, VPD indirectly influences RECO through its impact on GPP. The fact that the impact of VPD on the interannual variability of RECO is just as significant as on GPP supports this idea (see Figs. A6c and A7c).

Note that we restrained the number of climate variables in the revised manuscript for conciseness purpose. We now examine the climate impact using 3 variables: 2 m temperature, soil moisture and VPD. These variables exert a significant, physically grounded influence on NEE, GPP and RECO, justifying their inclusion in the analysis.

While multicollinearity poses a strong issue when analysing the combined influence of climate variables on $CO_2$ fluxes (e.g., in multiple linear regression), this study computes correlations variable by variable, which is methodologically sound and unaffected by multicollinearity.

5.Focus on Temperature: The authors only consider temperature when analyzing carbon fluxes in Figure 6 and do not include other important variables, such as soil moisture, which were emphasized in the introduction. Given this, the use of polynomial regressions without considering other factors raises questions about interactive effects of multivariate factors of the carbon fluxes.

The polynomial regressions presented in Figure 7 of the initial manuscript were useful for discussing temperature-induced threshold effects, which are known to influence both GPP and RECO. However, with the addition of new models and sites, retaining Figure 7 would have taken up too much space. Therefore, it has been removed in the revised manuscript.

Specific Points:

Please note that most of Reviewer 2's specific comments are no longer applicable, as the revised manuscript has been extensively rewritten.

Inconsistent Visualization (Figures 5 and 13): Figures 5 and 13 present similar data for annual and monthly scales, but the visualizations need to be consistent to enable direct comparison.

Figure 13 has been removed in the revised manuscript to maintain a strict focus on the monthly timescale and to further explore the climate–$CO_2$ flux relationship throughout the

annual cycle. Additionally, Figure 5 has been significantly revised to incorporate the 19 sites and the additional data included in the updated manuscript.

Climate Anomalies Definition: The authors should clarify how climate anomalies are defined, as the methods section only explains $CO_2$ flux anomalies. Also, the choice of the -0.5/+0.5 thresholds for carbon flux anomalies seems arbitrary and needs further justification.

This remark is no longer applicable, as the revised manuscript no longer includes results from composite analyses.

**Response to Community reader #1**

Section 1 Introduction:
Many of your references seem to be missing from this preprint or are cited incorrectly, e.g. Smith et al., 2020; Thompson et al., 2020; Yuan et al., 2009

Thank you for your remark. Many references were missing from the reference list due to a bug. We have carefully reviewed and updated the list to include all missing references.

Section 2.1 Site description:
It would be useful to know when each forest was established ( e.g. 10s,100s or 1000s of year ago.)

We completely agree that this information is valuable. However, we were unable to find it in the FLUXNET database or website. That said, this omission does not significantly impact our study, as (1) we do not analyze the effect of forest age, and (2) neither the data-driven nor process-based models explicitly account for forest age.

---

## Author Response (AR2)

**Response to reviewer's comments**

**Editor**
We have received two reviews. Please note that one referee was unable to review the manuscript a second time. A new (expert) reviewer was able to review the manuscript. Both referees agree the work was greatly improved with respect to the former comments. Referee #3 has a major comment that should be evaluated. I think there is value in the study's detailed comparison between model data and field observations at the interannual scale. However, they do have a point about which datasets we can draw physical insight from in interpreting these interannual results. Before publication, we recommend major revisions to address this point from Referee #3. Please also consider the other detailed points from both reviewers.

We sincerely thank the Editor for the opportunity to publish our study in *Biogeosciences*, and we are grateful to Referee #3 for agreeing to review our manuscript. The reviewer's suggestions offer valuable insights into the interannual variability of $CO_2$ fluxes. However, they diverge from the direction and focus emphasized by the first two reviewers. While we acknowledge the relevance of these comments, we believe they extend beyond the scope of our paper, which primarily aims to evaluate model performance and its complementarity with observational data and to demonstrate seasonal shifts in the relationship between climate and $CO_2$ fluxes at the interannual timescale.

Furthermore, the limited number of flux tower sites in European forests, combined with the relatively short duration of the available datasets, currently limits our capacity to fully address the reviewer's suggestions within the framework of this study. That said, we have made efforts to incorporate Referee #3's remarks, notably by expanding the discussion and rewording certain sections of the manuscript to better reflect these points.

Please find below our detailed, point-by-point responses to the reviewer's comments. Kindly note that the line numbers mentioned in our responses refer to the clean version of the manuscript (without track changes).

**Anonymous Referee #2 (Report #2)**
This is a second review report of the manuscript "Climate impact on mean annual cycle and interannual variability of CO2 fluxes in European DBF and ENF forests: insights from observations and state-of-the-art data-driven and process-based models". I found the authors have largely improved their work by adding comprehensive site measurements and data-driven and process-based products, as well as using those data at their best spatial resolution to more accurately compare with site data. The authors also implement multiple hydro-climate drivers in understanding the model behavior. I have a few more points and would like to suggest a publication after minor revision.

1. In Figure 3, why do all winter months in European sites have positive soil moisture, especially some months the temperature is below -5 or -10?

ERA5-Land soil moisture represents the total volumetric water content in the soil, including both liquid water and ice. This is why soil moisture values remain above zero even when temperatures fall below freezing. This clarification has been added in Section 2.2.4 (l281-283).

2. Can the authors give any hypothesis that why all products have systemic underestimated NEE interannual variability while except for CarbonSpace?

The potential reasons that could explain the better performance of the CarbonSpace model in capturing the observed NEE interannual variability are now discussed. It now reads (l620-627): "The interannual variability in the models is weaker than in the observations, consistent with previous studies (e.g., Nelson et al., 2024). However, the CarbonSpace data-driven model proved to be the only model tested that does not underestimate the NEE interannual variability. The reasons may involve its high spatial resolution (few hectares) and the use of a Lagrangian particle dispersion model, which allows it to closely align with the flux tower footprints. This results in more precise flux localization, which may improve its response to fine-scale variability. They may also involve the use of an ensemble tree method for regression. This method offers greater flexibility in capturing nonlinear interactions between environmental variables and NEE. Further studies are needed to evaluate these hypotheses".

3. How about these model performance in representing monthly anomalies when removing the seasonal cycles? This is also very relevant to guide product users and further model development.

Thank you for the suggestion. We added an analysis (new Fig. 9), which reveals weak impact of climate on NEE when all months are analyzed together. This apparent lack of influence arise from (and hides) distinct seasonal patterns. We modified Section 3.3.2 accordingly (l569-586).

**Anonymous Referee #3 (Report #1)**

In this revised manuscript, the authors have made substantial efforts to improve the original manuscript by incorporating most of the comments from previous reviewers. The key improvements are 1) shifting the research objective from evaluating models' performance to examining the climate impact on the mean annual cycle and interannual variability; 2) increasing the number of eddy covariance sites from three to 19; and 3) including more process-based and data-driven models. However, there is still one major issue regarding the interannual variability that I would like to see the authors address.

Thank you for taking time to review the two versions of the paper.

The major comment:

The authors have clearly shown that all process-based and data-driven products struggle to capture the magnitude and variance of the interannual variability of NEE (Figures 6 and 7).

This is a long-standing and known issue at least for data-driven products (Jung et al., 2020; Nelson et al., 2024). But they still claimed that these products can be used to analyze the climate impacts on the interannual variability of NEE. It is hard for me to follow this logic.

The rationale is that the magnitude of $CO_2$ flux interannual variability (as measured by the standard deviation) is largely independent from its temporal co-variability with climate (as measured by correlation). In other words, a model may exhibit a realistic magnitude of variability, while still failing to capture the correct climate-driven year-to-year fluctuations. Conversely, a model might underestimate (or exaggerate) the observed magnitude of interannual variability, but successfully capture the correct climate-driven year-to-year fluctuations. Therefore, we treat the magnitude (standard deviation) and temporal co-variability (correlation with climate) as distinct, uncorrelated aspects. It now reads:
- l311-313: "Two complementary metrics are used for model evaluation: the bias (model minus observation), which assesses errors in magnitude, and the Bravais-Pearson correlation coefficient (R), which evaluates temporal co-variability. These metrics capture distinct aspects of model performance and are not necessarily correlated."
- l463-465: "Importantly, a biased magnitude of $CO_2$ flux interannual variability (as measured by the standard deviation) does not preclude the models to capture their temporal co-variability (as measured by correlation) with observed $CO_2$ fluxes and climate."

Furthermore, models remain essential tools for assessing the relationship between climate and $CO_2$ fluxes at continental (e.g., Europe) to global scales. Although these models carry uncertainties related to input data, parameterization, and structural assumptions, they still provide valuable insights into the climate–$CO_2$ flux relationship. They are useful for identifying large-scale patterns, though they are not intended to accurately reproduce flux magnitudes or seasonal dynamics at specific sites. At the local scale, we acknowledge that direct climate measurements and site-specific observations offer more accurate information, provided that long-term data are available.

I agree with the authors that the length of some eddy covariance sites is probably not long enough to allow examining the interannual variability of NEE, but I do not think using long-term but problematic model products could help here. Highly inconsistent patterns among products in Figure 9 acknowledge this point.

This is true that patterns in Fig. 9 (now Fig. 10) are not identical across models, which is better acknowledge in the new version of the manuscript. While a single model cannot comprehensively capture the pattern, a model community (an ensemble of multiple models) at least capture the range of possible patterns and gives a common sense on patterns. In this study, several consistent features emerge between the multi-model ensemble and the observations. In winter and fall, NEE tends to be positively correlated with temperature, soil moisture, and VPD. In spring, NEE generally shows negative correlations with temperature and VPD, but a positive correlation with soil moisture. During summer, most models exhibit a negative correlation between NEE and soil moisture, as in the observations. Overall, the direction of the NEE–climate relationships is fairly consistent across the datasets.
We modified Section 3.3.2 to highlight common patterns and uncertainties (l569-586).

Nevertheless, examining the climate impact on the interannual variability of CO2 fluxes at the monthly scale is an interesting question and could improve our understanding of the ecosystem carbon cycling. Therefore, I would suggest that authors focus on eddy covariance data and completely give up using models, at least for the analysis of the interannual variability, which, of course, deviates a lot from the authors' initial idea, but would gain more scientific insights. In fact, the required length to represent the interannual variability is site-specific (Chu et al., 2017), therefore, authors could use the methods in Chu et al., 2017 to identify the sites with adequate temporal representativeness. If not enough sites are available, authors could extend the vegetation types beyond DBF and ENF, and/or extend the research area beyond Europe to increase the number of long sites.

This paper is part of a broader project aimed at evaluating the ability of models to capture $CO_2$ fluxes between European forest ecosystems and the atmosphere. In response to previous reviewer suggestions, we have made a huge effort to include additional data-driven and process-based models in this study. While uncertainties remain, the model results reveal clear and coherent signals for European forests (and uncertainties are a result *per se*). For these reasons, we have chosen to focus specifically on interannual variability within European forest ecosystems and do not extend the analysis to other ecosystems or regions.

To account for the reviewer remark, we have:
- modified the Abstract, Section 3.3.2 and the Conclusion;
- added a discussion regarding the representativeness of flux tower measurements and the need to disentangle site-specificities and regional signals to better calibrate model. It now reads:
  - l607-608: "Site-specific characteristics may also cause a disconnect between $CO_2$ flux variability and regional climate variability (Chu et al., 2017)."
  - l667-672: "The climate–NEE relationship is much noisier in both space and time in the observations than in the models, and it can vary substantially across different models. This indicates that local flux measurements may not reliably represent regional-scale dynamics, while models may exaggerate the influence of climate on $CO_2$ flux variability (despite underestimating its magnitude). Further work is needed to disentangle site-specific effects from broader-scale signals, a critical step toward improving the calibration of regional and global models that cannot resolve local heterogeneity".

Minor comments:

-Lines 71-78: These findings are questionable given the highly inconsistent patterns among products in Figure 9.

We don't fully agree with Referee #3 regarding the high inconsistency of the patterns. Our main point is that the influence of climate on $CO_2$ fluxes varies along the annual cycle. It now reads (l68-76): "At the interannual timescale, the climate does not show a significant influence on observed and modelled NEE when correlations are computed using monthly anomalies across all months combined. This apparent lack of relationship conceals meaningful seasonal patterns. In winter and fall, NEE tends to be positively correlated with temperature, soil moisture and VPD. In spring, NEE shows negative correlations with temperature and VPD, but

a positive correlation with soil moisture. The summer pattern is reversed compared to the spring pattern. In the observations, these relationships are noisy in both time and space, suggesting strong site-specific effects. In contrast, the models exhibit more structured and spatially coherent patterns with strong correlations, which may reflect an exaggerated response to climate forcing despite underestimated magnitude in $CO_2$ flux interannual variability".

-Figure 1: The labels of 'CH-Dav' and 'IT-Ren' overlap.

Done.

-Line3 165-171: The definition of Northern, central, and Southern Europe appeared here. But it is unclear to tell from Figures 3-9 themselves. Adding this information to these figures would improve the readability.

Thank you for this remark. We added this information to all Figures.

-Line 298: Incident shortwave radiation is an essential driver of NEE and GPP. I would suggest keeping it.

The aim of this paper is not to identify the dominant climate driver of $CO_2$ flux variability, but rather to demonstrate that the relationship between climate and $CO_2$ fluxes varies substantially from month to month at the interannual timescale. The current analysis, which focuses on 2 m temperature, soil moisture, and vapour pressure deficit, is sufficient to support this conclusion. This parameter shows no clear seasonality in correlation patterns with NEE, suggesting that greater light availability generally enhances $CO_2$ sequestration. For this reason, we chose not to include this variable in the present study. This is now stated in Section 2.2.4.

-Figure 2: If authors still want to keep the model data, this figure is not necessarily important and could go to the supplement.

We decided to retain Figure 2 as a main figure because it highlights key challenges in comparing observations and models, while also illustrating the potential of models to overcome the limited temporal coverage of observational data.

-Figure 3: Add the abbreviation 'IV' into the caption.

Done.

-Line 371: Any significance test was done here?

No test was performed. We rephrased as follows (l354-355): "While the interannual variability of $T_{2m}$ is the largest in winter regardless of the site, it increases markedly from south to north (Fig. 3d)".

-Lines 375-376: Why? Since it is not relevant, I would suggest removing it.

Removed as suggested.

-Lines 422-423: High correlation coefficient values could only indicate that the variance is captured well, but maybe not be true for the magnitude. Please rephrase it.

We rephrased as follows (l414-416): "The model skill in capturing the temporal phasing of the annual cycle and the magnitude in observed $CO_2$ fluxes is assessed in terms of correlation and mean bias, respectively (see section 2.3 for details). All models accurately capture the observed temporal phasing of GPP and RECO, with correlation values often above 0.8 (Fig. 5a)."

Note that correlation analyses are not performed for assessing magnitude errors, which is now clearly stated in section 2.3 (l311-313): "Two complementary metrics are used for model evaluation: the bias (model minus observation), which assesses errors in magnitude, and the Bravais-Pearson correlation coefficient (R), which evaluates temporal co-variability. These metrics capture distinct aspects of model performance and are not necessarily correlated".

-Lines 515-517: As mentioned above, this statement is questionable.

We removed this sentence, which was not necessary. However, we do not fully agree with Referee #3's point of view: despite the underestimated magnitude of interannual variability, the temporal co-variability can still be accurate (and vice versa).

-Lines 713-722: Same as above.

We maintained that the models are useful tools to complement observations for assessing the influence of climate on $CO_2$ flux interannual variability, but deleted speculative information. It now reads (l695-706): "Despite biased magnitude, the interannual variability of modelled fluxes correlates well with the observations. This supports the use of models to complement observations, whose limited temporal coverage and site specificities hinders the assessment of climate impacts on $CO_2$ interannual variability. We show that the influence of climate on $CO_2$ flux interannual variability is obscured when monthly anomalies are analyzed together. This apparent lack of relationship masks distinct seasonal patterns, which are concealed when considering all months together. Winter and fall $CO_2$ release increases under elevated temperature and VPD in northern and central Europe, while no clear signal emerges in southern Europe. The $CO_2$ sequestration increases under anomalously hot and dry conditions in spring and cold and wet conditions in summer in northern/central Europe. Anomalously

cold and wet conditions also favor $CO_2$ sequestration in southern Europe from spring to summer. While these seasonal signals appear noisy in the observations, due to limited sample sizes and site-specific variability, they emerge more clearly in the models, albeit with some model-dependent differences".